

# The EMEP Intensive Measurement Period campaign, 2008–2009: Characterizing the carbonaceous aerosol at nine rural sites in Europe

Karl Espen Yttri[*a], David Simpson[b,c], Robert Bergström[c,d], Gyula Kiss[e], Sönke Szidat[f], Darius Ceburnis[g], Sabine Eckhardt[a], Christoph Hueglin[h], Jacob Klenø Nøjgaard[i], Cinzia Perrino[j], Ignazio Pisso[a], Andre Stephan Henry Prevot[k], Jean-Philippe Putaud[l], Gerald Spindler[m], Milan Vana[n], Yan-Lin Zhang[jkm], Wenche Aas[a]

[a]NILU — Norwegian Institute for Air Research (NILU), N-2027 Kjeller, Norway

[b]Norwegian Meteorological Institute, 0313 Oslo, Norway

[c]Department of Space, Earth and Environment, Chalmers University of Technology, 41296 Gothenburg

[d]Swedish Meteorological and Hydrological Institute, 60176 Norrköping, Sweden

[e]MTA-PE Air Chemistry Research Group, 8200 Veszprém – Hungary

[f]Department of Chemistry and Biochemistry & Oeschger Centre for Climate Change Research, University of Bern, 3012 Berne, Switzerland

[g]School of Physics and Centre for Climate and Air Pollution Studies, Ryan Institute, National University of Ireland Galway, Galway, Ireland

[h]EMPA, CH-8600 Duebendorf, Switzerland

[i]National Environmental Research Institute, DK-4000 Roskilde, Denmark

[j]CNR — Institute of Atmospheric Pollution Research, 00015 Monterotondo Stazione (Rome), Italy

[k]Paul Scherrer Institut, 5232 Villigen-PSI, Switzerland

[l]European Commission, Joint Research Centre, I-21027 Ispra (VA), Italy

[m]Leibniz Institute for Tropospheric Research , 04318 Leipzig, Germany

[n]The Czech Hydrometeorological Institute (CHMI), Prague, Czech Republic

*To whom correspondence should be addressed: E-mail address: key@nilu.no



**Abstract**
Source apportionment (SA) of carbonaceous aerosol was performed as part of the EMEP Intensive
Measurement Periods (EIMPs), conducted in fall 2008 and winter/spring 2009. Levels of elemental
carbon (EC), particulate organic carbon ($OC_p$), particulate total carbon ($TC_p$), levoglucosan and $^{14}C$ in
$PM_{10}$, observed at nine European rural background sites, were used as input for the SA, whereas Latin
Hypercube Sampling (LHS) was used to statistically treat the multitude of possible combinations
resulting when ambient concentrations were combined with appropriate emission ratios. Five
predefined sources/subcategories of carbonaceous aerosol were apportioned: Elemental and organic
carbon from combustion of biomass ($EC_{bb}$ and $OC_{bb}$) and from fossil fuel ($EC_{ff}$ and $OC_{ff}$) sources, as
well as remaining non-fossil organic carbon ($OC_{rnf}$), typically dominated by natural sources.
The carbonaceous aerosol concentration decreased from South to North, as did the concentration
of the apportioned carbonaceous aerosol. $OC_{rnf}$ was more abundant in fall compared to winter/spring,
reflecting the vegetative season, and made a larger contribution to $TC_p$ than anthropogenic sources
(here: $EC_{bb}$, $OC_{bb}$, $EC_{ff}$ and $OC_{ff}$) at four of the sites, whereas anthropogenic sources dominated at all
but one sites in winter/spring. Levels of $OC_{bb}$ and $EC_{bb}$ were typically higher in winter/spring than in
fall, due to larger residential wood burning emissions in the heating season, whereas there was no
consistent seasonal pattern for fossil fuel emissions. Biomass burning ($OC_{bb}$ + $EC_{bb}$) was the major
anthropogenic source at the Central European sites in fall, whereas fossil fuel sources dominated at the
southernmost and the two northernmost sites. In winter/spring, biomass burning was the major
anthropogenic source at all but two sites. Addressing EC in particular, fossil fuel sources dominated at
all sites in fall, whereas there was as shift towards biomass burning in winter/spring for the
southernmost sites. Influence of residential wood burning emissions was substantial already in the first
week of sampling in fall, constituting 30 – 50% of $TC_p$ at most sites, showing that this source can be
dominating even at a time of the year when the ambient temperature in Europe is still rather high.
Model calculations were made, attempting to reproduce LHS-derived $OC_{bb}$ and $EC_{bb}$, using two
different residential wood burning emission inventories. Both simulations strongly under-predicted the
LHS-derived values at most sites outside Scandinavia. Emissions based on a consistent bottom-up
inventory for residential combustion (and including intermediate volatility compounds, IVOC)
improved model results at most sites compared to the base-case emissions (based mainly on officially
reported national emissions), but at the three southernmost sites the modelled $OC_{bb}$ and $EC_{bb}$
concentrations were still much lower than the LHS source apportioned results.
The current study shows that natural sources is a major contributor to the carbonaceous aerosol in
Europe even in fall and in winter/spring, and that residential wood burning emissions can be equally
large or larger than that of fossil fuel sources, depending on season and region. Our results suggest that
residential wood burning emissions are still poorly constrained for large parts of Europe. The need to
improve emission inventories is obvious, with harmonization of emission factors between countries
likely being the most important step to improve model calculations, not only for biomass burning
emissions, but for European $PM_{2.5}$ concentrations in general.



**Introduction**

Atmospheric aerosol particles play an important role in a number of environmental topics such as the radiation transfer of the Earth's atmosphere, the hydrological cycle as well as air quality, and thus have a substantial impact on the biosphere, including human health (Pope and Dockery, 2006; Andreae and Ramanathan, 2013). Carbonaceous matter is an important component of aerosol particles that has been found to account for 10–40% of $PM_{10}$ in the European rural background environment, 20–50% of $PM_{2.5}$ in urban and rural locations, and up to 70% of $PM_1$ (Zappoli et al., 1999; Putaud et al., 2010; Yttri et al., 2007a; Zhang et al., 2007; Querol et al., 2009). The carbonaceous matter is the least understood fraction of atmospheric aerosol particles due to its complexity in terms of composition, sources and formation mechanisms (Gelencsér, 2004; Pöschl, 2005; Hallquist et al., 2009; Ziemann 2012). Nevertheless, it is considered to have specific impacts on global climate (Novakov and Penner, 1993; Kanakidou et al., 2005), and on human health (Bell et al., 2009; Rohr and Wyzga, 2012; Cassee et al., 2013).

Particulate carbonaceous matter covers a wide range of organic components from low molecular weight hydrocarbons, through complex mixtures of humic-like substances and high molecular weight biopolymers containing also oxygen, nitrogen and sulphur, to tar balls or particles consisting of graphene layers. This continuum in chemical composition is reflected also in its thermochemical and optical properties (Pöschl et al., 2003). The carbonaceous fraction is usually quantified by its carbon content (total carbon, TC), which can be operationally divided into carbonate, organic carbon (OC), and elemental (EC) or black carbon (BC).

The complexity of carbonaceous aerosol originates from the diversity of its sources and formation processes. Carbonaceous particles are emitted both from anthropogenic (e.g. fossil fuel and biomass combustion) and biogenic sources (e.g. primary biological aerosol particles, PBAP, such as fungal spores, bacteria and degraded plant material). In addition to primary aerosol (emitted in particle form), carbonaceous aerosol can form by atmospheric oxidation of volatile precursors emitted by the vegetation or anthropogenic sources. Because of its climate forcing and adverse health effects, as well as its considerable contribution to particulate mass, source apportionment of carbonaceous aerosol is of key importance. By [14]C-analysis, carbonaceous aerosol from fossil and modern sources can be distinguished and quantified (Szidat et al., 2004; Szidat et al., 2009; Heal et al., 2011), and whereas fossil carbon is only emitted as a consequence of human activities, modern carbon originates from both biogenic and anthropogenic sources. Thus, source-specific tracers are necessary to apportion the modern carbon content. Levoglucosan, characteristic for wood burning emission, is the most commonly used macrotracer, whereas arabitol, mannitol and cellulose are used to distinguish different types of PBAP, another source of contemporary carbon. The combination of [14]C and source-specific organic tracer analysis has proved to be an efficient method for source apportionment of the carbonaceous aerosol (Gelencsér et al., 2006; Gilardoni et al., 2011; Yttri et al. 2011a, b; Liu et al., 2016). Studies combining [14]C- and [13]C-analysis for source apportionment are also reported (Ceburnis et al. 2011).

Globally, biomass burning is the major source of the carbonaceous aerosol (Crutzen and Andreae, 1990; Gelencsér, 2004), but the form and volume combusted (savanna fires, tropical forest fires,

agricultural waste burning, residential wood burning, etc.) depend highly on the geographical position,
climate and economic situation. In Europe, wood burning for residential heating, wild fires and
agricultural waste burning are the dominant forms of biomass burning, and thus significant sources of
carbonaceous aerosol, although these sources were hardly recognized for large parts of Europe, until
recently. Reviewing source apportionment studies of particulate matter in Europe between 1987 and
2007, Viana et al. (2008) stated that in spite of its importance at certain locations, biomass combustion
had rarely been identified as a substantial contributor to PM levels. Gelencsér et al. (2007) and May et
al. (2009) studied anthropogenic versus natural contribution to the total organic carbon content in
aerosol samples collected at six non-urban sites along a west-east transect over Europe from the Azores
(Portugal) to K-puszta (Hungary) and found biogenic sources to dominate at all sites in summer. In
winter most of the carbonaceous aerosol was emitted from anthropogenic sources, but there was a
considerable difference in the contribution of biomass burning and fossil fuel combustion, depending
on the geographical location (primarily altitude) of the sampling sites. Recently, a number of
measurement based studies have discussed the role of residential wood burning as a source of air
pollution in European urban and rural environments. As an example, road traffic and wood combustion
contributed equally to the annual mean $PM_{10}$ concentrations at various sites in Switzerland (Gianini et
al., 2012). In rural environment in the Alps, the contribution of wood burning to $PM_{10}$ even exceeded
that of road traffic (Gianini et al., 2012), and in Alpine valleys wood burning was the dominant source
of carbonaceous particles in wintertime (Szidat et al., 2007; Gilardoni et al., 2011; Herich et al., 2014;
Zotter et al., 2014). Similar results were found both in rural and urban environments in Norway by Yttri
et al. (2011a), who concluded that 80–90% of the winter time carbonaceous aerosol was emitted from
anthropogenic sources, and that wood burning contributed slightly more than fossil-fuel sources. In
summer, however, 70% of TC was attributed to natural sources in the rural environment, whereas the
corresponding number for the urban environment was 50%.

Modelling studies from recent years confirm that wood burning emissions are important in
wintertime Europe, and that such emissions seem to be severely underestimated in many regions
(Simpson et al., 2007; Bergström et al., 2012; Genberg et al., 2013). Denier van der Gon et al. (2015a)
pointed at inconsistent emission factors as a major problem (some countries report mainly solid
emissions, whereas others include substantial amounts of condensed semi-volatile OC, SVOC), and
produced a new bottom-up emission inventory for residential wood burning emissions of OC and EC,
using a consistent methodology across Europe (see also Genberg et al., 2013). Modelling work based
upon this inventory, and also including associated intermediate volatility compounds (IVOC),
improved model results for both EC and OC at European regional background sites (Genberg et al.,
2013 and Denier van der Gon et al., 2015a) but, so far, only limited comparisons to source
apportionment data have been made with model simulations using the new inventory.

The EMEP (European Measurement and Evaluation Program) task force on measurement and
modelling (TFMM) periodically arranges intensive measurement periods (IMPs), as a supplement to
the continuous monitoring in EMEP (Aas et al., 2012). The present study is part of the second EMEP
IMP, which was organized in cooperation with the EU-funded project EUCAARI (European Integrated
project on Aerosol, Cloud, Climate, and Air Quality Interactions: Kulmala et al., 2009; Crippa et al.,





2014) in fall 2008 and winter/spring 2009. In this study, collection of aerosol filter samples and
measurements of $^{14}$C, levoglucosan and OC/EC were harmonized by common protocol and analysis in
centralized laboratories. The objective was to provide quantitative estimates of carbonaceous aerosol
from fossil fuel, biomass burning and natural sources in the European rural background environment,
and to study their relative contribution in two transition periods, in which a noticeable signal from all
the considered sources was expected. The carbonaceous aerosol apportioned to biomass burning was
used to evaluate model simulated $EC_{bb}$ and $OC_{bb}$ with both a base-case emission inventory, based
mainly on official nationally reported emissions, and a recent, consistent, bottom-up estimate of
residential combustion emissions. In the current paper we present the main findings from our study.
**1.      Experimental**
**1.1      Site description and measurement period**
Aerosol filter samples were collected at nine European rural background sites (Table 1, Figure 1) for a
fall period (17 September–15 October 2008; denoted Fall) and a winter/spring period (25 February–25
March 2009; denoted Winter/spring). For a description of the sampling sites, see Appendix A.
**1.2      Aerosol sampling**
Ambient aerosol filter samples were obtained using various low volume filter samplers equipped with a
$PM_{10}$ inlet, collecting aerosol on prefired (850 °C; 3 h) quartz fiber filters (Whatman QMA; 47 mm in
diameter, batch number 11415138). The only exception was for samples collected at the Mace Head
station, which used a high volume sampler with a $PM_{2.5}$ inlet. The samplers were operated at a flow
rate ranging from 16.7 l min$^{-1}$ to 1.71 m$^3$ min$^{-1}$, corresponding to a filter face velocity ranging from 20
to 69 cm s$^{-1}$ (Table 1). The filter samples were collected according to the Quartz fiber filter behind
Quartz fiber filter (QBQ) approach to provide a quantitative estimate of the positive sampling artefact
of organic carbon (OC), thus the impact of the different filter face velocities at the various sites should
be minimized. The sampling time was one week, and four samples were collected at each site for each
of the two periods. At Mace Head, the collection of filter samples deviated slightly from the protocol in
Fall 2008, as the second week of sampling was divided into two to separate polluted air masses passing
over the European continent for the first three days of the week and clean marine air masses for the last
four days of the week. The sampling inlets were installed approximately 4 m above ground level,
except at Mace Head (10 m). Post exposure, filter samples were placed in petri-slides and stored in a
freezer (-18 °C) to prevent degradation or evaporation of the analytes.
**1.3      Thermal-optical analysis**
Total carbon (TC), elemental carbon (EC), and organic carbon (OC) were quantified using the Sunset
Lab OC-EC Aerosol Analyzer (Birch and Cary, 1996), using transmission for charring correction and
operated according to the EUSAAR-2 temperature program (Cavalli et al., 2010)



### 1.4 Determination of non-fossil TC from $^{14}$C analysis

For the measurement of $^{14}$C(TC$_p$) ($^{14}$C of particulate TC), 0.2–2 cm$^2$ punches, corresponding to 4–40 µg TC, were transferred into preheated quartz tubes (4 mm outer diameter) filled with ~0.1 g cupric oxide. The tubes were connected to a vacuum line, cooled to -70 °C, evacuated to <10$^{-3}$ hPa within one minute and then sealed. The sealed ampoules were heated to 850 °C for 4 hours for oxidation of TC to carbon dioxide (Fahrni et al., 2010). $^{14}$C measurements were performed at the Laboratory of Ion Beam Physics of ETH Zurich, using the accelerator mass spectrometer MICADAS, equipped with a gas ion source (Ruff et al., 2007), which allowed a direct injection of the carbon dioxide after dilution with helium (Wacker et al., 2013). $^{14}$C results for the front filters were corrected for SVOC contributions using the TC mass of the corresponding back filters and the mean $^{14}$C result of the four back filters for the respective site and season. $^{14}$C(TC$_p$) values are given as fractions modern (F$^{14}$C), i.e. as the $^{14}$C/$^{12}$C ratios of the samples related to the isotopic ratio of the reference year 1950 (Reimer et al., 2004). For determination of the non-fossil fraction of TC$_p$ (i.e., f$_{nf}$(TC$_p$) from $^{14}$C(TC$_p$) determinations, a reference F$^{14}$C value of pure non-fossil emissions of 1.08±0.04 was used to consider the different impacts of excess $^{14}$C from atmospheric nuclear bomb tests to fresh biomass and tree wood (Mohn et al., 2008). This is based on the assumptions that 50% of non-fossil TC originates from fresh biomass and 50% from burning of wood, whereof the latter includes 10-year, 20-year, 40-year, 70-year and 85-year old trees with weights of 0.2, 0.2, 0.4, 0.1, and 0.1, respectively.

### 1.5 Measurement of levoglucosan, mannosan and galactosan

Quantification of the monosaccharide anhydrides (MA) levoglucosan, mannosan and galactosan was performed according to the method described by Dye and Yttri (2005), which has been successfully applied for aerosol samples ranging from the urban (e.g. Fuller et al., 2014) to the remote environment (e.g. Yttri et al. 2014).

For the analysis, punches (1.5 cm$^2$) of the filter were spiked with $^{13}$C$_6$-levoglucosan and $^{13}$C$_6$-galactosan and extracted twice with 2 ml tetrahydrofuran under ultrasonic agitation (30 min). The filtered extracts (Teflon syringe filter, 0.45 µm) were evaporated to a total volume of 1 ml in a nitrogen atmosphere. Before analysis the sample solvent elution strength was adapted to the mobile phase by adding Milli-Q water (0.8 ml). The concentrations of the MAs were determined using High-Performance Liquid Chromatography (HPLC) (Agilent model 1100) in combination with HRMS-TOF (high resolution time-of-flight mass spectrometry, Micromass model LCT) operated in the negative ESI mode. Levoglucosan, mannosan and galactosan were identified on the basis of retention time and mass spectra of authentic standards. Quantification was performed using isotope labeled standards of levoglucosan and galactosan. The mass traces at $m/z$ 161.0455 and 167.0657 were used for quantification (approximately 50 mDa peak width).

The method described has been subject to intercomparison (Yttri et al., 2015).



### 1.6.1 Measurement uncertainties

### 1.6.1 Estimating the positive sampling artefact of OC

Table 2a and b show the $OC_{Back}/OC_{Front}$ ratios for the various sites. $OC_{Back}$ is gaseous OC present on the back filter and $OC_{Front}$ is the sum of gaseous and particulate OC on the front filter. This ratio provides an estimate of the magnitude of the positive sampling artefact (i.e. adsorption of semi volatile organic species on the filter/ collected particles) of OC when using tandem filter sampling. When subtracting $OC_{Back}$ from $OC_{Front}$, positive-artefact-corrected particulate organic carbon ($OC_p$) is obtained.

The positive artefact of OC ranged from 5.9±1.0 % (K-puszta, HU) to 28±13 % (Lille Valby, DK) in fall, whereas the corresponding range in winter/spring was 6.6±1.3 % (Ispra, IT) to 30±10 % (Lille Valby, DK). This shows that $OC_p$ could be severely overestimated if the positive artefact was not accounted for. Note that the QBQ approach does not account for any negative artefacts (i.e. release of semi volatile organic species from collected particles), thus the $OC_p$ levels should be considered as conservative estimates. There was typically a minor difference in the magnitude of the positive artefact between fall and winter/spring. No seasonal pattern consistent for all sites was observed.

### 1.6.2 Uncertainties in OC/EC measurements

~15 µg EC cm$^{-2}$ is considered the upper limit for the Sunset Lab OC-EC Aerosol Analyzer (Subramanian et al., 2006; Wallén et al., 2010), and should not be exceeded in order to obtain a correct OC/EC split. A non-biased OC/EC split also requires that either pyrolytic carbon (PC) evolves before EC or that PC and EC have the same light absorption coefficient, which we know is not always the case (Yang and Yu, 2002). In Fall 2008 11/36 samples exceeded 15 µg EC cm$^{-2}$, whereas the corresponding number for winter/spring 2009 was 3/36. For most of these samples the concentration just barely exceeded 15 µg EC cm$^{-2}$, nevertheless there is an added, non-quantifiable, uncertainty for these samples compared to those for which EC < 15 µg C cm$^{-2}$.

### 1.6.3 Uncertainties in levoglucosan analysis

Yttri et al. (2015) reported that the analytical method used to quantify levoglucosan in the current study had a bias of -13±4% compared to the assigned value, being the median value of levoglucosan based on the values reported by all participating laboratories in the actual intercomparison.

### 1.6.4 Uncertainties of the $f_{nf}(TC_p)$ determination from $^{14}C$ analysis

Uncertainties of $^{14}C(TC)$ measurements were 1–4% for the front filters and 2–10% for the pooled back filters. The uncertainties of the front filters increased upon calculation of $^{14}C(TC_p)$, especially for filters with high SVOC contributions. A further increase occurred when determining $f_{nf}(TC_p)$ ($f_{nf}$ = fraction non fossil) due to the uncertainty of the reference $f_M$ value of pure non-fossil emissions so that the final uncertainties of the non-fossil fraction of $TC_p$ given in Table 2a and b ranged from 0.03 to 0.09.

Two samples from Birkenes and two from Košetice had unrealistically high $^{14}C$ values, for unknown reasons. This finding was confirmed when rerunning the samples at another research institute. There are other examples showing that super modern carbon can be an issue for TC measured at European rural background sites (e.g. Glasius et al., 2018). Several hypothesis were suggested with





respect to what are the sources of super-modern carbon in the atmosphere: e.g. emissions from nuclear
power plants, waste incinerators taking care of waste from laboratories and hospitals, and crematoriums
(Buchholz et al., 2013; Zotter et al., 2014). Although samples highly contaminated with super-modern
$^{14}$C are easily observed, it is not possible to determine if reasonable looking samples are free from such
contamination. $^{14}$C contaminated measurements may lead to an overestimation of sources that emit
modern carbon when performing source apportionment of the carbonaceous aerosol, as described in the
current paper.

**1.7    Chemical transport modelling**
An important use of the carbonaceous aerosol Latin Hypercube Sampling (LHS) based source
apportionment, is to evaluate and constrain model systems for simulating particulate matter in the
atmosphere. The EMEP MSC-W model (Simpson et al., 2012; 2017 and references therein) is an Open
Source chemical transport model widely used for research, within the EMEP programme, and
elsewhere (e.g. Simpson et al., 2007; Bergström et al., 2012; 2014; Dore et al., 2015; Ots et al., 2016;
Vieno et al., 2016). In the present study we run the EMEP model with a horizontal resolution of 50 km
× 50 km across Europe, using 21 vertical levels, the lowest level being approximately 50 m thick.
Meteorological data from the Integrated Forecast System model (IFS; Cycle 40r1) of the European
Centre for Medium-Range Weather Forecasts (ECMWF) were used to drive the model. For this study,
version rv4.15 of the model was used with some modifications: The OC emissions from all sources
(except wildfires and open agricultural fires, which were treated as non-volatile for simplicity) were
treated as semi-volatile, and subject to evaporation and oxidation in the gas-phase (ageing), using a
volatility basis set (VBS) approach, similar to the VBS PAA scheme in Bergström et al. (2012; the
PAA scheme includes gas-particle Partitioning of primary organic aerosol emissions and Aging of All
semi-volatile OA components in the gas-phase). The model was run for the years 2008 and 2009, with
two different emission set-ups (See Sects. 2.7.1.1 and 2.7.1.2) in order to evaluate model performance
for biomass-burning derived OC and EC with these inventories.

**1.7.1    Emissions**
European residential wood burning inventories have substantial inconsistencies between countries
(Denier van der Gon, 2015a; Simpson and Denier van der Gon, 2015), and several assumptions
concerning volatility and oxidation-processes for such emissions are possible (e.g. Robinson et al.,
2007; Grieshop et al., 2009; Bergström et al., 2012; May et al. 2013a; Jathar et al., 2014; Ciarelli et al.,
2017). To illustrate some of the uncertainties associated with this, two different emission set-ups were
applied in the present study: A base-case run using the widely used MACC-III emission inventory, and
an alternative run, denoted DT+IVOC.

In both cases, anthropogenic emissions (except as noted below) were based on the TNO

MACC emission inventory for 2011 (Kuenen et al., 2014; Denier van der Gon et al., 2015b) with
emission categories following the SNAP system, in which SNAP-2 includes non-industrial combustion,
such as residential wood burning. Emissions from vegetation fires and agricultural burning were taken


from the Fire INventory from NCAR version 1.5 (FINNv1.5; Wiedinmyer et al., 2014) and OC
emissions from these types of fires were treated as non-volatile.

**1.7.1.1 Base Case**
For SNAP-2, the MACC-III emissions were split into biomass burning sources (mainly wood and
woody fuels) and fossil fuel sources (coal, oil etc.), using data from Kuenen (pers. comm., 2017). The
emissions in MACC-III were split into five volatility bins, with saturation concentrations ($C^{*}_{298K}$, in the
range 0.01–1000 µg m$^{-3}$) as shown in Table 3.

**1.7.1.2 *DT+IVOC Case***
POA and EC SNAP-2 emissions from MACC-III were scaled (except for Russia, for which the
MACC_III emissions were used also in the DT+IVOC runs) to better match the bottom-up inventory
'DT' from Denier van der Gon (2015a), where DT refers to data from dilution tunnels, which capture
condensables (SVOC) in addition to solid particles. This causes a substantial increase in POA
emissions for some countries (e.g. by more than a factor of three for Germany), but only minor for
others (e.g. Norway), as discussed by Denier van der Gon, (2015a). The DT/IVOC case adds extra
emissions of intermediate volatility compounds (IVOC) for all primary OA (POA) sources, as in
Denier can der Gon (2015a). The split between biomass burning (non-fossil) emissions and fossil fuel
based emissions for SNAP-2 was taken from the inventory of Denier van der Gon (2015a). Table 3
details the volatility assumptions used for the DT+IVOC case. EC emissions from wood combustion
are also different in the two different inventories (see Genberg et al., 2013, for a detailed discussion of
the EC emissions in the DT emission inventory).

**2.     Source apportionment using Latin Hypercube Sampling**
Source apportionment of TC into different source categories of fossil-fuel, biomass burning and
remaining non-fossil carbon for OC and EC has been done with chemical and $^{14}$C tracers. This
methodology, which is very similar to that used in Yttri et al. (2011a), was originally developed for the
CARBOSOL project (Gelencsér et al., 2007), and has been refined over the years, and applied in
several Nordic studies (Szidat et al., 2009, Yttri et al., 2011a, b, Glasius et al., 2018). In summary:
Measurements of levoglucosan are used as a tracer of wood-burning emissions ($TC_{bb} = OC_{bb} + EC_{bb}$;
$OC_{bb}$ includes primary and secondary OC) and the $^{14}$C isotopic ratio ($F^{14}C$), along with measured OC
and EC, and assumed emission ratios (e.g. $TC_{bb}$/levoglucosan and $OC_{bb}/TC_{bb}$ from wood combustion,
or OC/EC ratios from fossil-fuel combustion), to assign the remaining carbon between fossil-fuel
sources and secondary organic aerosol sources. When available (as in Yttri et al., 2011a), mannitol and
cellulose can be used as tracers of primary biological aerosol particles ($OC_{PBAP}$) derived from fungal
spores ($OC_{pbs}$) and plant debris ($OC_{pbc}$), respectively. Total carbon is in this way split into $TC_{bb}$,
$OC_{PBAP}$, $TC_{ff}$ (= $OC_{ff} + EC_{ff}$, from fossil-fuel sources; $OC_{ff}$ includes primary and secondary OC), and
finally, any remaining modern-carbon is labeled $OC_{rnf}$, which typically is dominated by $OC_{BSOA}$
(biogenic secondary organic aerosol), but might also include other sources, such as SOA from biomass
burning and emissions related to cooking (Mohr et al., 2009; Crippa et al., 2014). Note that Crippa et





al. (2014) did not find any influence of cooking at European rural background sites doing a source apportionment study of the carbonaceous aerosol based on Aerosol Mass Spectrometer (AMS) measurements. The relationship between any tracer and its derived TC component is very uncertain, thus an uncertainty distribution of allowed parameter values for all important emission ratios or measurement inputs is assigned. In order to solve the system of equations, allowing for the multitude of possible combinations of parameters, an effective statistical approach known as Latin-hypercube sampling is used, which is comparable to Monte Carlo calculations. In brief, central values with low and high limits are associated to all uncertain input parameters. These factors are combined using LHS in order to generate thousands of solutions for the source-apportionment. All valid combinations of parameters (i.e. excluding those producing negative solutions) are condensed in frequency distributions of possible solutions. Extensive discussion of the choices behind the factors used, and their uncertainties, can be found in earlier related studies (Yttri et al., 2011a, Szidat et al., 2009 Gelencsér et al., 2007, Simpson et al., 2007). The result of this analysis consist of so-called central-estimates of the TC components (i.e. the 50th percentile), as well as the range of possibilities allowed by the LHS calculation, e.g. expressed as the 10th and 90th percentiles of the solutions.

There are two major differences in the data available for this study compared to Yttri et al. (2011a, b), requiring modification of the methodology and factors used: i) For the present study, we have no data to estimate the fractions of PBAP and BSOA, thus $OC_{rnf}$ comprises both $OC_{BSOA}$, $OC_{PBAP}$ and indeed all other non-fossil sources of OC; ii) The geographical scope of the current study is wider, and in particular biomass burning in southern Europe involves different tree species than those used in the Northern European studies of Yttri et al. (2011a,b) or Szidat et al. (2009).

Concerning item (i), we require a range of values of the $F^{14}C$ value associated with $OC_{rnf}$. In Yttri et al. (2011a,b) we used 1.055 for BSOA and PBAP associated with plant debris, but allowed $F^{14}C$ for spores to vary between 1.055 and 1.25, reflecting the utilization of older carbon-stocks by fungi. As noted above, we have no direct tracers for BSOA or PBAP, but a few studies allow a general estimate. Winiwarter et al. (2009) suggested that fungal spores were likely the dominant contributor to PBAP across Europe. Results scaled for Europe indicated a contribution of PBAPs to $PM_{10}$ concentrations in the low percentage range, with a maximum in summer when $PM_{10}$ concentration levels are small. Similarly, Bauer et al. (2008) had spores contributing 6% to OC in spring and 14% in summer at a suburban site, whereas the corresponding contribution to $PM_{10}$ was 3% (spring) and 7% (summer). In Norway, Yttri et al. (2011a) found spores and debris contributing 18% and 6%, respectively, to TC at a rural site in summer, with 0.5% and 7% respectively in winter. For comparison, BSOA contributed 56% and 11% of TC in summer and winter at the actual site. Hence, spores and plant debris are likely to make a certain contribution, but are unlikely to dominate $OC_{nf}$. In order to account for this, we allow $F^{14}C$ to vary between 1.055 to 1.100 in the present study.

Concerning item (ii), the main effect is likely to be on the assumed TC/levoglucosan ratios used in the LHS method. In Yttri et al. (2011a,b) we used low, central and high values of 11, 15 and 17 for $PM_{10}$, or 7.6, 12, and 14 for $PM_{2.5}$, factors derived from ambient Norwegian data, and modified to be appropriate to the QBQ sampling used for the LHS. These values also seem to be consistent with the study of Elsasser et al. (2012), which reported OC/levoglucosan values from filter samples of about



10–17 for Augsburg, Germany. Inclusion of EC would give $TC_{bb}$/levoglucosan values at the high end
of our assumed range.
We have no equivalent data for southern Europe, but a simple examination of the data in Table
2 suggests that levoglucosan levels can be high at the Italian sites, and assuming high ratios of
(TC/levoglucosan)$_{bb}$ in emissions would result in LHS-estimated $TC_{bb}$ higher than observed TC, which
clearly is impossible. Gilardoni et al., (2011) used (OC/levoglucosan)$_{bb}$ of 4 to 13, then (OC/EC)$_{bb}$ of 1
to 20, whereas Zotter et al. (2014) observed (OC/levoglucosan)$_{bb}$ of 7.8±2.7 and (OC/EC)$_{bb}$ of 8.6±2.9
for Southern Switzerland, which is close to the Italian site Ispra. It isn't obvious how to derive
(TC/levoglucosan)$_{bb}$ from these values, but low values are clearly suggested by these choices.
In order to allow for this possibility, we have extended the lower range of our
(TC/levoglucosan)$_{bb}$ ratio to be 5, thus using low, central and high of 5, 15 and 17 for $PM_{10}$. This
actually made very little difference to the LHS solutions for central and northern Europe, but allowed
more solutions for the Italian sites.
No attempts to run LHS were possible for samples with unrealistically high $^{14}C(TC)$ values,
affecting two samples each from Birkenes and Košetice. No valid solution was obtained for five of the
samples collected at Ispra, two at Melpitz, one at Birkenes and one at Payerne. This may be an
indication of problems with the samples (e.g. artefacts or contaminated $^{14}C(TC)$ values), or with the
assumptions underlying LHS breaking down. Nevertheless, LHS-based source apportionment was
obtained for 29/35 samples in fall and for 29/36 in winter/spring.

**3.     Results**
**3.1     Ambient concentrations of the carbonaceous aerosol**
Concentrations of elemental carbon (EC), positive-artefact-corrected particulate organic carbon (OC$_p$),
organic carbon on back filters (OC$_B$), positive-artefact-corrected particulate total carbon (TC$_p$) and
levoglucosan, as well as the EC/TC$_p$ ratio and the $f_{nf}(TC_p)$ fraction observed during the fall 2008 and
the winter/spring 2009 intensive measurement periods, are presented in Table 2.

**3.1.1     EC and OC$_p$**
The mean EC concentration (0.64±0.58 µg C m$^{-3}$ in fall; 0.58±0.50 µg C m$^{-3}$ in winter/spring) was
quite similar to the annual mean concentration reported for 12 European rural background (EMEP)
sites in 2002–2003 (0.66±0.39 µg m$^{-3}$; Yttri et al., 2007a), but slightly less than the winter time mean
(0.79±0.83 µg C m$^{-3}$; ibid.). Although thermal-optical analysis was used both in the present study and
in that by Yttri et al. (2007a), different temperature protocols can cause substantial differences in the
OC/EC split. However, only a minor difference was observed with respect to the EC/TC ratio when
analyzing the "8785 Air Particulate Matter On Filter Media" reference material from NIST using the
EUSAAR-2 protocol and the NIOSH derived protocol (Yttri et al., 2007a). The mean EC concentration
varied by a factor of ~15 between sites both in fall and in winter/spring, with concentrations at
Birkenes and Mace Head (North-western Europe) being substantially lower than for continental
European sites, particularly compared to the southern sites (Montelibretti, Ispra and K-puszta). A
pronounced North-to-South gradient for EC, and OC, has previously been reported by Yttri et al.





(2007a), reflecting diluted emissions from major source regions in continental Europe reaching distant and less polluted sites on the outskirts of Europe. In addition, the proximity to the coast causes efficient ventilation and air mass mixing at the sites Birkenes and Mace Head.

The mean $OC_p$ concentrations in fall ($2.9\pm3.1$ µg C m$^{-3}$) and winter/spring ($2.8\pm2.3$ µg C m$^{-3}$) were almost identical. A few, high concentration samples at the sites Montelibretti, Ispra and K-puszta influenced the winter/spring mean, as evident from the mean-to-median ratio of 1.6 compared to 1.2 in fall. Mean $OC_p$ concentrations reported here were slightly lower than the annual ($3.4\pm3.6$ µg C m$^{-3}$) and winter time ($3.7\pm4.4$ µg C m$^{-3}$) mean OC concentrations reported for EMEP sites in 2002−2003 (Yttri et al., 2007a). Differences in sampling time, temperature protocol, and sampling approach [the current study accounted for the positive sampling artefact of OC, whereas Yttri et al., (2007) did not], are likely to explain the (minor) differences in the OC concentration between the two studies. If we allow for a positive artefact of similar magnitude as that observed in the present study, $16\pm8$ % in fall and $17\pm9$ % in winter/spring, also for the Yttri et al. (2007a) study, levels would be fairly similar.

A North-to-South gradient was observed for $OC_p$ as for EC, which was less prominent in fall compared to winter/spring.

### 3.1.2    EC/TC ratio

The EC/$TC_p$ ratio ranged from 11 to 28 % in fall, and from 14 to 24 % in winter/spring. No pronounced shift in the EC/$TC_p$ ratio was observed between the two periods, except for the Norwegian site Birkenes, for which the EC/$TC_p$ ratio was 11% in fall and 21% in winter/spring.

### 3.1.3    Levoglucosan

The mean concentration of the wood burning tracer levoglucosan varied by more than a factor of 50 between sites, both in fall and in winter/spring. There was a pronounced North-to-South gradient, as for $OC_p$ and EC and the mean concentration was higher in winter/spring than in fall at all sites, except Košetice and Mace Head. The levoglucosan level is within the range reported for six European rural background sites ($2.7–1220$ ng m$^{-3}$) by Puxbaum et al. (2007), and for Montelibretti, Ispra, and K-puszta, levels equaled the concentration range reported for urban areas in winter (Szidat et al., 2009).



### 3.1.4    $f_{nf}(TC_p)$ from [14]C analysis

The non-fossil fraction of $TC_p$ (i.e. $f_{nf}(TC_p)$) of individual aerosol filter samples varied from 0.51 to >1.00. Two samples from Birkenes and two samples from Košetice showed such high [14]C(TC) results that the corresponding $f_{nf}(TC_p)$ resulted in levels as high as 1.68. These unreasonable values point to an anthropogenic bias of local [14]C emissions, which distort the source apportionment. Similar cases have occasionally been observed at other sites, mainly caused by local pharmaceutical facilities with incineration units for [14]C-labelled waste (Buchholz et al., 2013; Zotter et al., 2014). In some cases, the specific source could not be identified, as for Birkenes and Košetice. Consequently, the biased values were excluded from further analysis. The remaining results from these two sites were included, as they correspond well with values from the other sites, although their reliability remains unclear.

Mean $f_{nf}(TC_p)$ values ranged from 0.61–0.91 for the individual sites, including both fall and winter/spring. These values correspond to those reported at five European rural and remote sites in summer and winter by Gelencsér et al. (2007) and to an urban and a rural site in Norway (Yttri et al., 2011a), but are higher compared to rural and urban sites in Switzerland and Sweden during summer and winter (Szidat et al., 2009). The seasonal variation was typically not pronounced, although most sites experienced the highest $f_{nf}(TC_p)$ values in winter/spring. The exceptions were Montelibretti, at which $f_{nf}(TC_p)$ was noticeably higher in winter/spring (0.80) compared to fall (0.61), and Košetice at which $f_{nf}(TC_p)$ was higher in fall 2008 (0.86) compared to winter/spring 2009 (0.69).

## 4.    Discussion

Results from the carbonaceous aerosol source apportionment (Figure 2; Table 4) show a variability in the carbonaceous aerosol source composition, both as a function of season and location. The results from the source apportionment analyses are discussed in detail in sections 5.1–5.5. Calculated concentrations and relative contributions typically showed little variability between samples collected within each season for each of the nine sites. Hence, comparing results based on calculated mean values can be argued for. The results presented are complementary to those of Gelencsér et al. (2007), Genberg et al. (2011) and Yttri et al. (2011a,b), as the same same (or similar in the case of Genberg et al.) software/methodology is applied, but for a wider range of sites, and with updated emission ratios (Zotter et al., 2014) for the central and southern European sites.

### 4.1    Carbonaceous aerosol from fossil-fuel sources and biomass burning

Fossil fuel combustion was the major source of EC at all sites in fall, accounting for 6% to 22% of $TC_p$, whereas EC from biomass burning was < 8% at all sites. The influence of $EC_{ff}$ was particularly pronounced at the sites Montelibretti (22%) and Lille Valby (21%), which for Montelibretti could be due to the proximity of the Rome metropolitan area, with 3.7 million inhabitants. Lille Valby is a semi-rural site, and thus could be more influenced by e.g. vehicular particulate emissions. Fossil fuel combustion continued to be the most important source of EC in winter/spring for the five northernmost sites, whereas there was a shift towards biomass burning for the four southernmost sites. The relative contribution of $EC_{bb}$ and $EC_{ff}$ to $TC_p$ in winter/spring was $\leq$ 10%, except at the sites Lille Valby, Melpitz and Birkenes that experienced relative contributions of $EC_{ff}$ exceeding 10%. $EC_{bb}$ was a more



abundant fraction of $TC_p$ in winter/spring compared to fall at all sites. The picture was less consistent
for $EC_{ff}$, with a higher relative contribution in fall at the four southernmost sites, and for Lille Valby,
and a higher fraction in winter/spring for the four other sites.

Biomass burning was the major anthropogenic source of OC at most sites in fall, accounting

from 5% to 36% of $TC_p$, whereas OC from fossil fuel ranged from 8% to 21%. The exceptions were
Birkenes and Mace Head for which $OC_{ff}$ dominated with 16% and 21%, respectively. At Montelibretti,
$OC_{bb}$ and $OC_{ff}$ made equally large contributions to $TC_p$ (18% each).

In winter/spring, biomass burning was the major anthropogenic source of OC at all sites

except at Mace Head, constituting 11% to 46% of $TC_p$, whereas the range for $OC_{ff}$ was 10% to 23%.
$OC_{bb}$ was more abundant in winter/spring compared to fall for all sites but Mace Head, whereas there
was no consistent pattern observed for $OC_{ff}$. There was a general tendency that $OC_{bb}$ became less
abundant along a South-to-North transect, as seen for $EC_{bb}$.

Biomass burning had a pronounced influence at most sites already in the first week of

sampling in fall (17–24 September): $EC_{bb}$ and $OC_{bb}$ contributed a substantial 57% of $TC_p$ at K-puszta
and 54% at Ispra, 34% and 37% at Melpitz and Payerne, respectively, whereas it ranged from 21–29%
for the sites Mace Head, Košetice and Lille Valby. Birkenes was the only sites where wood burning
made a minor contribution (6%) this week. Model calculations suggest that wild and agricultural fires
were of minor importance at all sites for the actual week, with the highest model calculated
concentration (0.02 µg C m$^{-3}$) at Ispra and Lille Valby, corresponding to 3% and 5% of the modelled
$TC_{bb}$ (See section 5.2). Hence, residential wood burning appears to be the source of $EC_{bb}$ and $OC_{bb}$,
although given the uncertainties of emission estimates for wild and agricultural fires, such sources
cannot be ruled out. The mean temperature during the first week of sampling was not noticeably lower
than seen for the rest of the sampling period. Still, it was the week with the lowest mean temperature
for the sites K-puszta, Payerne and Košetice.

**4.2 Wild and agricultural fire contribution**
Wild and agricultural fires are major sources of carbonaceous aerosol (Bond et al., 2004), but with
large regional, seasonal and annual differences in emissions and occurrence (Hao et al., 2016; Korontzi
et al., 2006). Agricultural waste burning is banned in most European countries, as it is a major source
of forest fires, and thus a threat to human life and properties, as well as a source of severe air pollution.
Nevertheless, remote sensing data show such fire events in several countries, including those with a
ban (Korontzi et al., 2006), and it appears particularly frequent in Eastern Europe (e.g. Belarus and the
Ukraine), in western parts of Russia and in Central Asia. In most cases when natural vegetation catches
fire in Europe, this is due to human activity (Winiwarter et al., 1999).

Incidences of wild and agricultural fires that severely deteriorate air quality in large parts of

Europe are regularly reported e.g. by Yttri et al. (2007a) for 2002, by Stohl et al. (2007) for 2006, and
Diapouli et al. (2014) for 2010. The two periods discussed in the present study partly coincide with the
time when concentrations from wild and agricultural fires peak in Europe (Korontzi et al., 2006).
Levoglucosan by itself cannot differentiate between emissions from residential wood burning and wild
and agricultural fires. Hence, we have used modelled concentrations to address the relative contribution



of TC from wild fires and agricultural fires ($TC_{wf}$) to the sum of TC from residential wood burning
($TC_{bb}$) and $TC_{wf}$ for the two sampling periods.
There was an influence from wild and agricultural fires at all sites, with a higher mean
contribution in fall ($TC_{wf}$ = 0.05 µg C m$^{-3}$), corresponding to 9–16% (for base-case, or DT+IVOC) of
modelled $TC_{bb}$, than in winter/spring ($TC_{wf}$ = 0.015 µg C m$^{-3}$), corresponding to 2–4% of modelled
$TC_{bb}$. $TC_{wf}$ were typically low also on a weekly basis, but for the last week of sampling in fall, a
noticeable contribution was calculated for Ispra (34%), K-puszta (31%), and Montelibretti (16%).
The major conclusion to be drawn from these results is that the model predicts that wild and
agricultural fires make minor contributions to the biomass burning carbonaceous aerosol at the sites
addressed, and that residential wood burning is the major source.

**4.3      Remaining non-fossil sources of organic carbon**
Remaining non-fossil sources of OC ($OC_{rnf}$) are typically associated with biogenic secondary organic
aerosol ($OC_{BSOA}$) and primary biological aerosol particles ($OC_{PBAP}$), however there are anthropogenic
sources of modern carbon as well, as discussed in detail by Yttri et al. (2011a). Here, we discuss the
results obtained for $OC_{rnf}$ as if natural sources are dominating.
The $OC_{rnf}$ level varied more widely in winter (0.1–2.2 µg C m$^{-3}$) than in fall (0.6–3.0 µg C m$^{-3}$)
(Figure 2) and corresponds well with levels reported for the European rural background environment
(Gelencsér et al., 2007; Genberg et al., 2011; Yttri et al., 2011a,b). The spatial distribution of $OC_{rnf}$
equaled that of $OC_p$, with high concentrations at the southernmost sites and decreasing levels along a
South-to-North transect.
$OC_{rnf}$ levels were higher in fall compared to winter/spring for all sites, but the difference
varied from minor at most sites, moderate at the continental sites Košetice and Payerne, and substantial
at the Norwegian site Birkenes. Studies consistently point towards BSOA as the major contributor to
$OC_{rnf}$ in Europe (e.g., Simpson et al., 2007; Bessagnet et al., 2008; Yttri et al., 2011a); e.g. Gelencsér et
al. (2007) showed that BSOA in $PM_{2.5}$ was 1.6–12 times higher in summer than in winter for six
European rural background sites. Hence, the observed pattern could partly be explained by a higher
formation rate of BSOA in fall, propelled by larger emissions of BSOA precursors and a higher
ambient temperature (See Table 1 ambient temperature values). In the present study, $PM_{10}$ filter
samples were collected (except at Mace Head, where $PM_{2.5}$ was collected). Consequently, primary
biological aerosol particles (PBAP), typically residing in the coarse fraction of $PM_{10}$ (e.g., Yttri et al.,
2007b; Kourtchev et al., 2009; Bozzetti et al., 2016), could contribute to $OC_{rnf}$ as well. In Scandinavia,
PBAP peak in summer and fall, reflecting the vegetative season and the absence/presence of a snow
cover (Yttri et al., 2007a,b; 2011a,b), and summer time $OC_{PBAP}$ concentrations ($PM_{10}$) being 7–8 times
higher than in winter, has been reported for two Norwegian sites (Yttri et al., 2011a). In continental
Europe, the vegetative season is longer than in Scandinavia and a permanent snow cover is associated
with high altitude regions and rare occasions, lasting for short periods, in low altitude regions. Hence,
one could speculate that there is a PBAP emission flux in continental Europe in the heating season,
which is comparatively larger than that observed in Scandinavia. We find support of this view in the
study by Waked et al. (2014), which showed a tail of PBAP and episodes with high PBAP



concentrations in winter for an urban background site in Northern France. Knowledge of PBAP
concentrations in Europe is limited, thus we can only speculate about how much of $OC_{rnf}$ in the present
study is due to PBAP. A noticeable 20–32% contribution of $OC_{PBAP}$ to $TC_p$ was found at four Nordic
rural background sites in late summer (Yttri et al., 2011b). Similar figures (OC from primary biogenics
constituting up to 33% of OC in $PM_{10}$) were reported for the densely populated region of Berlin in
north-eastern Germany (Wagener et al., 2012) in late summer and fall. Gelencsér et al. (2007) and
Gilardoni et al. (2011) both reported levels of OC associated with PBAP for an entire year for the
European rural background environment, finding that the relative contribution to total carbon was < 5%
in summer and < 8% in winter. However, both studies relied on $PM_{2.5}$ samples, likely excluding the
majority of PBAP. Further, Gelencsér et al. (2007) accounted for plant debris only when measuring
cellulose, whereas Gilardoni et al. (2011) only accounted for fungal spores, measuring
arabitol/mannitol. Waked et al. (2014) found that 17% of the OC was attributed to $OC_{PBAP}$ on an annual
basis for an urban background site, with substantially higher concentrations in summer (37%) and fall
(20%) compared to winter (7%) and spring (6%). At the rural background site Payerne, Bozzetti et al.
(2016) found that PBAP, mainly from plant debris, equaled the contribution of SOA to organic matter
in $PM_{10}$ in summer.

The non-fossil signal was typically most pronounced in fall, with the highest relative share (52 –

69%) observed for the two low loading sites situated on the outskirts of Europe (Birkenes and Mace
Head) and the lowest for the highest loading site, Ispra (23%). A pronounced non-fossil signal (52 –
54%) was seen for the continental sites Košetice and Payerne as well, whereas the relative share ranged
between 38% and 48% for the remaining sites. Non-fossil OC was by far the major source of OC at all
sites in fall, except at Ispra, for which biomass burning dominated. The non-fossil signal decreased, or
remained unchanged, for all but one site going from fall to winter/spring, but the reduction was
substantial only at the Norwegian site Birkenes (a factor of ~2), at Payerne and Košetice (a factor of
1.5–1.7), and at Melpitz (a factor of 1.5). Still, non-fossil OC was the major source of OC at five sites
even in winter/spring, K-puszta, Košetice, Lille Valby, Mace Head and Birkenes. It has been suggested
that increased condensation due to lower temperatures could be an efficient way of forming BSOA
even in winter (Simpson et al., 2007). It is however difficult to argue for such a hypothesis barely by
looking at the observed ambient air temperatures during the winter/spring period. Another possibility is
that some of the remaining non-fossil OC may be secondary organic aerosol formed from volatile or
semi-volatile OC emitted from wood burning. $OC_{bb}$ determined based on levoglucosan may not include
all SOA formed after aging of the gas-phase emissions, even if the emission ratios were derived from
ambient measurements and likely include condensed vapors and secondary products.

**4.4    Natural versus anthropogenic sources of carbonaceous aerosol**
In the current study, results obtained for $OC_{rnf}$ are discussed as if natural sources are dominating,
despite that anthropogenic sources can make a certain contribution, e.g. from cooking emissions and by
anthropogenic enhancement of BSOA formation. EC and OC emitted from combustion of fossil fuel
and biomass are considered entirely anthropogenic, as we define wild fires as anthropogenic.



In fall, the anthropogenic and natural influences were of comparable magnitude at most sites.
Exceptions were Birkenes, with a clearly larger natural contribution (69%), and Ispra, with a larger
anthropogenic contribution (77%), the latter affected by regional air pollution in the strongly polluted
Po Valley region. For the other sites, the anthropogenic fraction ranged from 46 – 62% and from 38 –
54% for the natural fraction. Increased condensation due to lower temperatures can be an important
source of BSOA in fall and winter, which could outweigh the effect of high temperature and increased
terpene emissions in summer (Simpson et al., 2007). Further, PBAP can make a pronounced
contribution in fall both in Scandinavia (Yttri et al., 2007a,b; 2011a,b) and in continental Europe
(Waked et al., 2014; Bozzetti et al., 2016), and the fall peak of the North-Eastern Atlantic Ocean
phytoalgal bloom takes place during the period in question, likely contributing with marine PBAP at
Mace Head (Ceburnis et al., 2011).
In winter/spring, anthropogenic sources dominated at all sites (60 – 78% anthropogenic),
except for Mace Head (37%). Ispra had the most pronounced anthropogenic contribution of all sites
also in winter/spring (78%), and it was largely unchanged from that observed in fall. Three of the four
sites experiencing a high natural influence in fall, (Birkenes, Košetice and Payerne) saw a major
increase in the anthropogenic contribution going from fall to winter/spring. This was attributed to a
substantial reduction in natural sources, accompanied by an increase in the anthropogenic sources,
being primarily biomass burning at Payerne and Birkenes and fossil fuel sources at Košetice.
Residential wood burning is considered a decentralized source in Europe, and emissions from local
sources can be substantial in winter (Szidat et al., 2007). A certain local contribution could also be
speculated for Košetice, as small coal-fired ovens still are common in rural areas in Eastern Europe
(Spindler et al., 2012).

**4.5 Modelling contributions from biomass burning**
The EMEP MSC-W model was run with two different emission and SOA modelling set-ups (a base-
case and DT+IVOC) in order to reflect (to some extent) the very large uncertainties in both emissions
and atmospheric processing of the primary organic aerosol (POA) (see section 2.7). The model results
were compared with that of the LHS analysis discussed above. In the following, model results that are
within the 10–90 percentile range of the LHS analysis are considered as being in "agreement" with the
measurements. Results outside this (fairly wide) concentration range are considered as under or over
estimations.
Modelled $OC_{bb}$ and $EC_{bb}$ concentrations were compared to the LHS source apportionment
results for each sample individually in Figure 3, and as averages over the measurement periods in Table
4. The base-case model simulations underestimated $OC_{bb}$ severely at most sites (Figure 3a). The only
exception was Birkenes, for which the model slightly overestimated the LHS-derived estimates (the
modelled $OC_{bb}$ were within the LHS 10–90 percentile range for 3/5 weeks, whereas 2/5 weeks were
overestimated). For the other sites, the mean underestimation of the LHS 10-percentile for $OC_{bb}$ ranged
from −26% at Lille Valby to −84% at Payerne.
The model results for $OC_{bb}$ were clearly better with the DT+IVOC emission set-up (Figure
3b), than for the base-case, at all sites except Birkenes and Lille Valby. For Košetice and Payerne, the





modelled $OC_{bb}$ was within the LHS range for a majority of the samples and the underestimation of
$OC_{bb}$ was smaller than with the base-case for Ispra, Montelibretti, K-puszta and Melpitz. A few
individual $OC_{bb}$ measurements were, however, clearly overestimated with the DT+IVOC setup (one
sample each for Melpitz, K-puszta and Lille Valby).
The results for $EC_{bb}$ roughly split in two groups for the base-case (Figure 3c): At Birkenes and
Lille Valby, the $EC_{bb}$ concentrations were overestimated by the model most of the time; only for one
sample at each site did the model $EC_{bb}$ fall within the LHS-range. The average overestimation of the
LHS 90-percentile was 69% at Lille Valby and 43% at Birkenes. At the other sites, $EC_{bb}$ was
underestimated (with a few exceptions), with an average underestimation ranging from −34%
compared to the LHS 10-percentile at Melpitz to −84% at Mace Head. For the two Italian sites the
average underestimation was −38%, whereas it was −39% at K-puszta and Košetice and −60% at
Payerne.
The DT+IVOC model results were clearly better for $EC_{bb}$, except for the Italian sites and K-
puszta where the $EC_{bb}$ underestimation was larger due to lower
emissions in the inventory of Denier van der Gon et al. (2015a). $EC_{bb}$ was largely overestimated at the
Scandinavian sites, but not as much as for the base-case emissions. The modelled $EC_{bb}$ was within the
10–90 percentile LHS range for five of the weeks at Košetice and Payerne using the DT+IVOC
emissions, but there was still a tendency that levels were underestimated (one week was underestimated
at Košetice, two at Payerne). For Melpitz the modelled $EC_{bb}$ was within the LHS range for 3/6 weeks
(two weeks were underestimated and one overestimated).
The present comparison of modelled and LHS-derived biomass burning carbonaceous aerosol
concentrations, indicates that the base-case setup with the TNO MACC-III emission inventory, which
is similar to official EMEP $PM_{2.5}$ emissions estimates, likely underestimates emissions from residential
wood burning substantially in large parts of Europe. This is in line with the findings of Denier van der
Gon (2015a), and reflects that emissions are established following national practice that is inconsistent
between countries. Note that the inventory POA emissions were distributed across different volatility
classes for the DT+IVOC emissions, as for a typical VBS treatment, whereas we did not add IVOC to
the MACC-III emissions in our base-case. Although the DT+IVOC emission setup with updated wood
burning emissions and extra IVOC improved the model results, large uncertainties still remain, and it
cannot be excluded that wood burning emissions in some parts of Europe may be considerably larger
than that estimated by Denier van der Gon et al. (2015a).

**5    Conclusions**
Source apportionment of carbonaceous aerosol was conducted at nine European rural background sites
for a fall period in 2008 and a winter/spring period in 2009. The approach separated the carbonaceous
aerosol into a natural and an anthropogenic fraction, and divided the anthropogenic fraction into fossil
fuel and biomass burning origin, which is a prerequisite for targeted abatement strategies. The fraction
apportioned to biomass burning was compared with calculated concentrations using the EMEP model,
applying a base-case and an alternative emission set up with intermediate volatility compounds
(IVOC).



689    The total carbonaceous aerosol concentration, as well as the carbonaceous aerosol apportioned
690 to biomass burning, fossil fuel and natural sources, decreased from South to North. Natural sources
691 typically accounted for a larger fraction of the carbonaceous aerosol in fall compared to winter/spring,
692 likely because the fall sampling period partly took place in the vegetative season. The seasonal
693 differences of the natural sources varied from minor at most sites, moderate at two of the continental
694 sites, to substantial at the northernmost Scandinavian site. Biomass burning aerosol had an opposite
695 seasonal behavior to that of natural sources, following the increased emissions from residential wood
696 burning in the heating season. No consistent seasonal pattern was observed for fossil fuel aerosol and
697 their contribution to the carbonaceous aerosol, possibly because domestic heating is a minor source of
698 fossil fuel carbon compared to e.g. vehicular traffic.

699    Anthropogenic sources (60–78%) dominated at all but the most remote site in winter/spring,
700 and residential wood burning (36–56%) was typically the major anthropogenic source of TC. In fall,
701 anthropogenic and natural influence were of comparable magnitude at most sites, except at Birkenes
702 (69% natural) and Ispra (77% anthropogenic). Biomass burning was the major anthropogenic source at
703 Central European sites in fall (29–44%), whereas fossil fuel dominated at the southernmost (40%) and
704 the three northernmost sites (29–37%).

705    Model calculated concentrations of carbonaceous aerosol from biomass burning were severely
706 underestimated, except for the Scandinavian sites, when using the base-case MACC-III emission
707 inventory. Model results improved when an alternative bottom-up approach with added IVOC was
708 used. However, $OC_{bb}$ and $EC_{bb}$ levels were still substantially underestimated at the southernmost sites.

709    The current study shows that natural sources are major contributors to the carbonaceous
710 aerosol at background sites in Europe even in fall and in winter/spring, and that residential wood
711 burning emissions can be equally large or larger than that of fossil fuel sources, depending on season
712 and region. Our results suggest that residential wood burning emissions are poorly constrained for large
713 parts of Europe and that the need to improve emission inventories is obvious, with harmonized
714 emission factors between countries likely being the most important step to improve model calculations.
715 Revised wood burning emissions will also improve model predictions of $PM_{2.5}$ concentrations in
716 Europe, particularly in the heating season. EMEP intensive measurement periods are essential for real-
717 world evaluation of model results, especially when the underlying emission data are so uncertain; as is
718 future EMEP intensive measurement periods targeted on the wood burning source.

719

720 *Author Contributions.* KEY was responsible for the main design, coordination of the study, the
721 synthesis of the results, writing most of the paper, responsible for the centralized analysis of
722 levoglucosan, and provide OC/EC data for Birkenes. DS did the Latin Hybercube Sampling (LHS), as
723 well as the EMEP modelling part together with RB. DS wrote the text on LHS, whereas DS and RB
724 together wrote the text on the modelling, as well as they thoroughly reviewed the paper. GK wrote the
725 introduction, provided OC/EC data for K-puszta and wrote the description of the site, and thoroughly
726 reviewed the paper. SS and Y-LZ were responsible for and performed the centralized [14]C-analysis,
727 wrote the text on this topic, and thoroughly reviewed the paper. WAA and ASHP contributed to the
728 coordination of the study and thoroughly reviewed the paper. CH provided OC/EC data for Payerne,





wrote the description of the site and thoroughly reviewed the paper. CP provided OC/EC data for
Montelibretti, wrote the description of the site and thoroughly reviewed the paper. DC provided OC/EC
data for Mace Head, wrote the description of the site and thoroughly reviewed the paper. GS provided
OC/EC data for Melpitz, wrote the description of the site and thoroughly reviewed the paper. JPP
provided OC/EC data for Ispra, wrote the description of the site and thoroughly reviewed the paper.
JKN provided OC/EC data for Lille Valby and wrote the description of the site. MV provided OC/EC
data for Košetice and wrote the description of the site. SE and IP thoroughly reviewed the paper.

*Competing interests.* The authors have no conflict of interest.

*Acknowledgements.* This work was supported by the Co-operative Programme for Monitoring and
Evaluation of the Long-range Transmission of Air pollutants in Europe (EMEP) under UNECE, the
European Union Seventh Framework Programme (FP7/2007–2013) under the ACTRIS project (Grant
agreement #262254), and the European Union Seventh Framework Programme (FP7/2007–2013)
under the ECLIPSE project Grant agreement #282688 – ECLIPSE. Computer time for EMEP model
runs was supported by the Research Council of Norway through the NOTUR project EMEP
(NN2890K), and this work was also supported by the Swedish Strategic Research Project MERGE
(www.merge.lund.se). We are grateful to the Laboratory of Ion Beam Physics of ETH Zurich for
providing the accelerator mass spectrometer MICADAS for $^{14}$C measurements. We thank ECMWF and
met.no for granting access to ECMWF analysis data. Hugo Denier van der Gon and Jeroen Kuenen
from TNO are acknowledged for useful discussions and data concerning OM emissions.




## APPENDIX A

**Detailed description of measurement sites**

The Montelibretti EMEP station is situated in central Italy (42°06'N, 12°38'E, 48 m asl) 45 km from the coast of the Thyrrenian sea. Most of the land surrounding the station are meadows and low intensity agricultural areas. The nearest village (Monterotondo, 30 000 inhabitants) is situated approximately 5 km from the station, whereas the City of Rome lies 20 km to the south-west. Transport of air masses from the urban area of Rome is typically associated with sea-breeze taking place in the early afternoon.

The Ispra station (45° 49'N, 8° 38'E, 209 m asl) is situated on the edge of the Po Valley in the north-western part of Italy and is representative for the regional background of this densely populated part of Italy. Major anthropogenic emission sources are situated > 10 km from the site, with the city of Milan, 60 km to the south-east, as the most pronounced one. According to Henne et al. (2010), Ispra is categorized as a typical background site in an environment generally strongly affected by anthropogenic emissions.

The Payerne measurement station (46°48'N, 6°56'E, 489 m asl) is part of the Swiss national air pollution monitoring network as well as the EMEP monitoring network, and is regarded as a rural site. The station is located one kilometre south-east of the small town of Payerne (8 000 inhabitants). The site is surrounded by agricultural land (grassland and crops), forests and small villages. The nearest larger cities are Fribourg (15 km east, 35 000 inhabitants), Bern (40 km north east, 125 000 inhabitants) and Lausanne (40 km south-west, 120 000 inhabitants).

The K-puszta station (46°58'N, 19°33'E, 130 m asl) is situated in a forest clearing on the Great Hungarian Plain and is representative for the Central-Eastern European regional background environment. The vegetation is dominated by coniferous wood (60%), but also deciduous wood (30%) and grassland are present. The nearest city (Kecskemét) is situated ca 15 km to the SE of K-puszta. The station is part of the Global Atmospheric Watch (GAW) network, the European Monitoring and Evaluation Programme (EMEP) and is also a EUSAAR supersite. The climate is typically continental with low temperatures in winter, mild in spring and fall, and hot and sunny in summer.

The Košetice observatory (49º35'N, 15º05'E, 534 m asl) is a joint EMEP and GAW site located in the Czech-Moravian Highlands, approximately 80 km southeast from Prague. Air samples collected at the observatory represents the background level of air quality in the Czech Republic. Forests dominated by conifer trees account for approximately 50% of the land use in the vicinity of the site; the remaining 50% is attributed to meadow (25%) and agricultural areas (25%). The nearest city (Pelhřimov, 15 000 inhabitants) is located 25 km south of the station. The prevailing wind direction is westerly.

The Melpitz research station (51°32' N, 12°54' E, 87 m asl) is located on a flat meadow surrounded by agricultural land near the river Elbe. The major city Leipzig is situated 41 km to the south west of the site. Forested areas are located no closer than 1 km from the site. The two dominating wind directions are south west to west, which brings air masses from the Atlantic that passes across Western Europe, and east to south-east, which brings air masses from source regions such as Poland, Belarus, Ukraine and the north of the Czech Republic.





The Mace Head atmospheric research station (53º19'N, 9º53'W, 15 m asl) is a GAW supersite
situated on the west coast of Ireland, facing the North Atlantic Ocean. The station is located 100 m
from the coastline and is surrounded by bare land (rocks, grass and peat bog). A few scattered single
houses are located at a distance of 1 km or further away. The nearest city (Galway, 80 000 inhabitants)
is located 60 km to the east/south-east of the station. The site experience clean marine air masses from
the western sector nearly 50% of the time, whereas polluted air masses are associated with atmospheric
transport from UK and continental Europe.
Lille Valby (55°41' N, 12°07' E, 12 m asl) is a semi-rural monitoring station in the Sjælland
region of Denmark, which has a humid continental climate. The surrounding area is characterized by
agricultural land, small villages and the Roskilde Fjord (1 km west of the monitoring site). The station
is located 30 km to the west of Copenhagen (1.2 million inhabitants), and 7 km North-East of central
Roskilde (46 000 inhabitants). The nearest major road (A6) is located about 800 m west of the station.
The Birkenes atmospheric research station (58°23'N, 8°15'E, 190 m asl) is a joint supersite
for EMEP and GAW situated approximately 20 km from the Skagerrak coast in southern Norway. The
station is located in the boreal forest with mixed conifer and deciduous trees accounting for 65% of the
land use in the vicinity of the site; the remaining 35% is attributed to meadow (10%), low intensity
agricultural areas (10%), and freshwater lakes (15%). The nearest city (Kristiansand, 65 000
inhabitants) is located 25 km south/south-west of the station, and is known to have minor or even
negligible influence on the air quality at the site.



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





**Table 1: Location of the nine European rural background sites that participated in the Fall 2008 and Winter/spring 2009 sampling periods. The sites are ordered by latitude from south to north.**

| Sampling site | Location | Height (m asl) | Sampling period | Cut-off size | Flow rate (l min$^{-1}$) | Filter face velocity (cm s$^{-1}$) | Ambient temp. (min–max) | Precip. (min–max) |
|---|---|---|---|---|---|---|---|---|
| Montelibretti (Italy) | 42° 06'N, 12° 38'E | 48 | 24.09–15.10.2008 | PM$_{10}$ | 38 | 54 | 16.8 (16.2-17.1) | 0.8 (0-2.4) |
| | | | 25.02–25.03.2009 | | | | 9.9 (8.5-11) | 16.6 (1.2-45.8) |
| Ispra (Italy) | 45° 48'N, 08° 38'E | 209 | 24.09–22.10.2008 | PM$_{10}$ | 16.7 | 20 | 13.0 (12.8-13.3) | NA |
| | | | 25.02–25.03.2009 | | | | 8.0 (7-9.6) | NA |
| Payerne (Switzerland) | 46° 48'N, 06° 56'E | 489 | 16.09–16.10.2008 | PM$_{10}$ | 16.7 | 23 | 10.5 (9.2-12.5) | 1.4 (0.6-2.5) |
| | | | 27.02–25.03.2009 | | | | 4.4 (2.9-6.5) | 1.4 (0-3.9) |
| K-puszta (Hungary) | 46°58'N, 19°33'E | 130 | 17.09–15.10.2008 | PM$_{10}$ | 16.7 | 22 | 11.7 (9.9-12.6) | 9.3 (0-19.4) |
| | | | 25.02–25.03.2009 | | | | 5.1 (3.7-7.2) | 5.3 (1.3-10.5) |
| Košetice (Czech Rep.) | 49°35'N, 15°05'E | 534 | 17.09–15.10.2008 | PM$_{10}$ | 38 | 53 | 9.6 (7.5-11.9) | 7.4 (2.7-16.6) |
| | | | 25.02–25.03.2009 | | | | 2.0 (0.4-3.4) | 17.3 (11.3-23.2) |
| Melpitz (Germany) | 51°32' N, 12°54'E | 87 | 17.09–15.10.2008 | PM$_{10}$ | 16.7 | 22 | 11.2 (10.6-12.3) | 7.6 (3.1-14.3) |
| | | | 25.02–25.03.2009 | | | | 5.4 (3.7-6.8) | 13.2 (9.5-16.6) |
| Mace Head (Ireland) | 53° 19'N, 09° 53'W | 15 | 18.09–15.10.2008 | PM$_{2.5}$ | 1111 | 45 | 12.4 (11.3–12.9) | 17.3 (0–51.2) |
| | | | 25.02–25.03.2009 | | | | 8.3 (7.1–9.4) | 12.4 (0.1–37.1) |
| Lille Valby (Denmark) | 55° 41'N, 12° 08'E | 10 | 17.09–15.09.2008 | PM$_{10}$ | 38 | 56 | 10.9 (9.2-12) | 7.6 (0.3-21.7) |
| | | | 25.02–25.03.2009 | | | | 5.2 (2.7-10.3) | 9.7 (3.3 21.3) |
| Birkenes (Norway) | 58° 23'N, 8° 15'E | 190 | 17.09–15.10.2008 | PM$_{10}$ | 38 | 54 | 8.2 (6-9.4) | 31.1 (7.6-53.1) |
| | | | 25.02–25.03.2009 | | | | -0.7 (-1.5-0.3) | 22.5 (0.2-48.5) |



**Table 2a: Mean (± SD) concentrations of carbonaceous sub-fractions and levoglucosan in PM$_{10}$[1] during Winter/Spring 2009. The EC/TC$_p$ ratio, the OC$_{Back}$/OC$_{Front}$ ratio and non-fossil fractions of TC$_p$ (f$_{nf}$(TC$_p$)) are also listed. The sites are ordered by latitude from south to north.**

| | Montelibretti | Ispra | Payerne | K-puszta | Košetice | Melpitz | Mace Head[1] | Lille Valby | Birkenes |
|---|---|---|---|---|---|---|---|---|---|
| *Unit: (µg C m$^{-3}$)* | | | | | | | | | |
| TC$_p$ | 6.1 ± 2.7 | 9.3 ± 5.7 | 3.6 ± 1.3 | 5.5 ± 2.8 | 2.1 ± 0.78 | 1.7 ± 0.68 | 0.76 ± 0.91 | 1.5 ± 0.33 | 0.44 ± 0.13 |
| OC$_p$ | 5.0 ± 2.5 | 7.9 ± 5.0 | 2.9 ± 1.0 | 4.8 ± 2.6 | 1.8 ± 0.70 | 1.3 ± 0.50 | 0.65 ± 0.79 | 1.2 ± 0.3 | 0.34 ± 0.08 |
| OC$_{Back}$ | 0.62 ± 0.16 | 0.50 ± 0.22 | 0.41 ± 0.18 | 0.35 ± 0.10 | 0.23 ± 0.09 | 0.41 ± 0.26 | 0.07 ± 0.04 | 0.53 ± 0.31 | 0.13 ± 0.13 |
| EC | 1.0 ± 0.25 | 1.5 ± 0.68 | 0.66 ± 0.27 | 0.77 ± 0.21 | 0.32 ± 0.12 | 0.40 ± 0.12 | 0.11 ± 0.13 | 0.37 ± 0.09 | 0.10 ± 0.05 |
| *Unit: (%)* | | | | | | | | | |
| EC/TC$_p$ | 18 ± 3.6 | 17 ± 2.3 | 19 ± 2.9 | 15 ± 3.3 | 16 ± 1.4 | 24 ± 4.1 | 14 ± 1.3 | 24 ± 5.4 | 21 ± 5.2 |
| OC$_{Back}$/OC$_{Front}$ | 12 ± 2.9 | 6.6 ± 1.3 | 12 ± 1.9 | 7.3 ± 1.4 | 12 ± 4.4 | 24 ± 12 | 23 ± 21 | 30 ± 10 | 24 ± 13 |
| *Unit: (Fraction)* | | | | | | | | | |
| f$_{nf}$(TC$_p$) | 0.80 ± 0.06 | 0.80 ± 0.05 | 0.90 ± 0.09 | 0.83 ± 0.09 | 0.69 ± 0.04 | 0.83 ± 0.13 | 0.79 ± 0.11 | 0.71 ± 0.13 | 0.77 ± 0.09 |
| *Unit: (ng m$^{-3}$)* | | | | | | | | | |
| Levoglucosan | 247 ± 113 | 668 ± 295 | 141 ± 63 | 209 ± 156 | 67 ± 16 | 57 ± 20 | 12 ± 13 | 41 ± 5.5 | 17 ± 7.7 |

1) For Mace Head PM$_{2.5}$ was used



**Table 2b:** Mean (± SD) concentrations of carbonaceous sub-fractions and levoglucosan in $PM_{10}$[1] during Fall 2008. The $EC/TC_p$ ratio, the $OC_{Back}/OC_{Front}$ ratio and non-fossil fractions of $TC_p$ ($f_{nf}(TC_p)$) are also listed. The sites are ordered from by latitude south to north.

| | Montelibretti[2] | Ispra | Payerne | K-puszta | Košetice | Melpitz | Mace Head[1] | Lille Valby | Birkenes |
|---|---|---|---|---|---|---|---|---|---|
| *Unit: ($\mu g\ C\ m^{-3}$)* | | | | | | | | | |
| $TC_p$ | 5.0 ± 1.8 | 7.6 ± 2.5 | 3.9 ± 1.1 | 6.7 ± 2.9 | 3.3 ± 0.66 | 2.1 ± 0.36 | 0.89 ± 1.2 | 1.8 ± 0.74 | 1.1 ± 0.47 |
| $OC_p$ | 4.0 ± 1.8 | 6.1 ± 2.0 | 3.3 ± 0.93 | 5.5 ± 2.7 | 2.8 ± 0.59 | 1.6 ± 0.21 | 0.77 ± 1.1 | 1.3 ± 0.70 | 0.97 ± 0.45 |
| $OC_{Back}$ | 0.75 ± 0.16 | 0.47 ± 0.31 | 0.53 ± 0.37 | 0.33 ± 0.08 | 0.21 ± 0.08 | 0.60 ± 0.33 | 0.10 ± 0.07 | 0.48 ± 0.21 | 0.17 ± 0.03 |
| EC | 0.97 ± 0.25 | 1.5 ± 0.54 | 0.59 ± 0.17 | 1.2 ± 0.26 | 0.49 ± 0.10 | 0.54 ± 0.16 | 0.12 ± 0.17 | 0.46 ± 0.10 | 0.11 ± 0.03 |
| *Unit: (%)* | | | | | | | | | |
| $EC/TC_p$ | 21 ± 8.3 | 20 ± 3.7 | 15 ± 0.31 | 18 ± 4.0 | 15 ± 2.1 | 25 ± 3.7 | 12 ± 5.6 | 28 ± 8.1 | 11 ± 3.3 |
| $OC_{Back}/OC_{Front}$ | 17 ± 3.8 | 6.8 ± 2.6 | 13 ± 4.9 | 5.9 ± 1.0 | 6.9 ± 1.5 | 26 ± 10 | 19 ± 8.9 | 28 ± 13 | 19 ± 6.7 |
| *Unit: (Fraction)* | | | | | | | | | |
| $f_{nf}(TC_p)$ | 0.61 ± 0.01 | 0.69 ± 0.08 | 0.80 ± 0.06 | 0.81 ± 0.03 | 0.86 ± 0.10 | 0.76 ± 0.04 | 0.70 ± 0.18 | 0.72 ± 0.12 | 0.75 ± 0.05 |
| *Unit: ($ng\ m^{-3}$)* | | | | | | | | | |
| Levoglucosan | 106 ± 40 | 364 ± 180 | 85 ± 16 | 172 ± 84 | 83 ± 14 | 33 ± 14 | 16 ± 19 | 32 ± 19 | 6.8 ± 2.2 |

1) For Mace Head $PM_{2.5}$ was used.

2) The sampler at Montelibretti was run in an alternating on/off mode, collecting ambient air 15 minutes every 1 hour.



**Table 3: Volatility distributions of the primary organic aerosol (POA) emissions from anthropogenic sources.**

| C* (µg m$^{-3}$)[a] | | $10^{-2}$ | $10^{-1}$ | 1 | 10 | $10^2$ | $10^3$ | $10^4$ | $10^5$ | $10^6$ |
|---|---|---|---|---|---|---|---|---|---|---|
| **Base-case emission fraction[b]** | SNAP 2 | 0.20 | 0.00 | 0.10 | 0.10 | 0.20 | 0.40 | 0.00 | 0.00 | 0.00 |
| | all other sources | 0.00 | 0.04 | 0.25 | 0.37 | 0.23 | 0.11 | 0.00 | 0.00 | 0.00 |
| **DT+IVOC emission fraction[c, d]** | SNAP 2 | 0.025 | 0.050 | 0.076 | 0.118 | 0.151 | 0.252 | 0.336 | 0.42 | 0.672 |
| | all other sources | 0.03 | 0.06 | 0.09 | 0.14 | 0.18 | 0.30 | 0.40 | 0.50 | 0.80 |

[a] C$^*$: Saturation concentration at 298 K; enthalpies of vaporization were taken from May et al. (2013a,b) for the base-case (MACC-III), and from Shrivastava et al. (2008) for the DT+IVOC case.

[b] The volatility distribution in the MACC-III model run is based on the recommended volatility distributions from May et al. (2013a,b) for biomass burning emissions (for SNAP sector 2; non-industrial stationary combustion) and for diesel exhaust (for all the other emission sectors), but moving the emissions in the C$^*$=$10^4$ µg m$^{-3}$–$10^6$ µg m$^{-3}$ bins to the $10^3$ µg m$^{-3}$ bin.

[c] The volatility distributions in the DT+IVOC case are based on Shrivastava et al. (2008) for all emission sectors except SNAP-2, for which it is based on the distribution used for the EMEP model in Denier van der Gon et al. (2015a). Note that this scenario assumes that there are substantial IVOC emissions that are not included in the emission inventories (see Bergström et al., 2012, and Denier van der Gon et al., 2015a).

[d] Since the DT emission inventory by Denier van der Gon et al. (2015a) was constructed to include a larger fraction of SVOC from residential wood burning emissions, we apply a slightly different emission split for the SNAP-2 POA compared to other SNAP sectors. Considering both SVOC and IVOC within the POA class, the total POA emissions are assumed to be 2.1 times the inventory (compared to the factor 2.5 for the other emission sectors).





**Table 4: Model and source apportioned (LHS-derived) concentrations of elemental carbon (EC$_{bb}$) and organic carbon (OC$_{bb}$) from biomass burning. Model results are averages over both measurement periods (Fall 2008 and Winter/Spring 2009). For the LHS-results the mean of the 10- and 90-percentiles are shown. Unit: µg C m$^{-3}$.**

| Site | EC$_{bb}$ | | | | OC$_{bb}$ | | | |
|---|---|---|---|---|---|---|---|---|
| | Base-case | DT+IVOC | LHS-10 | LHS-90 | Base-case | DT+IVOC | LHS-10 | LHS-90 |
| Montelibretti | 0.19 | 0.097 | 0.29 | 0.70 | 0.28 | 0.37 | 1.04 | 2.38 |
| Ispra | 0.34 | 0.21 | 0.47 | 0.93 | 0.63 | 0.82 | 1.70 | 3.16 |
| K-puszta | 0.20 | 0.17 | 0.30 | 0.67 | 0.37 | 0.74 | 1.10 | 2.27 |
| Payerne | 0.081 | 0.24 | 0.20 | 0.46 | 0.12 | 0.79 | 0.73 | 1.51 |
| Košetice | 0.074 | 0.17 | 0.12 | 0.28 | 0.14 | 0.60 | 0.42 | 0.91 |
| Melpitz | 0.063 | 0.096 | 0.085 | 0.18 | 0.12 | 0.37 | 0.30 | 0.57 |
| Mace Head | 0.0045 | 0.0091 | 0.028 | 0.057 | 0.015 | 0.061 | 0.086 | 0.16 |
| Lille Valby | 0.24 | 0.18 | 0.067 | 0.14 | 0.22 | 0.36 | 0.24 | 0.46 |
| Birkenes | 0.065 | 0.047 | 0.020 | 0.046 | 0.13 | 0.17 | 0.072 | 0.15 |



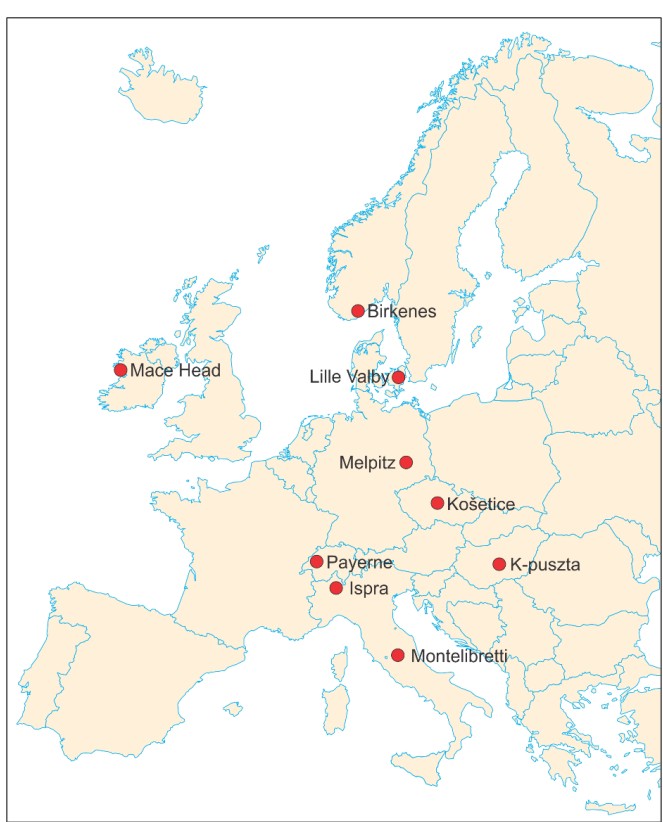

**Figure 1: Overview of sampling sites participating in the carbonaceous aerosol source-apportionment study in the EMEP intensive measurement periods (IMPs) in Fall 2008 and Winter/spring 2009.**





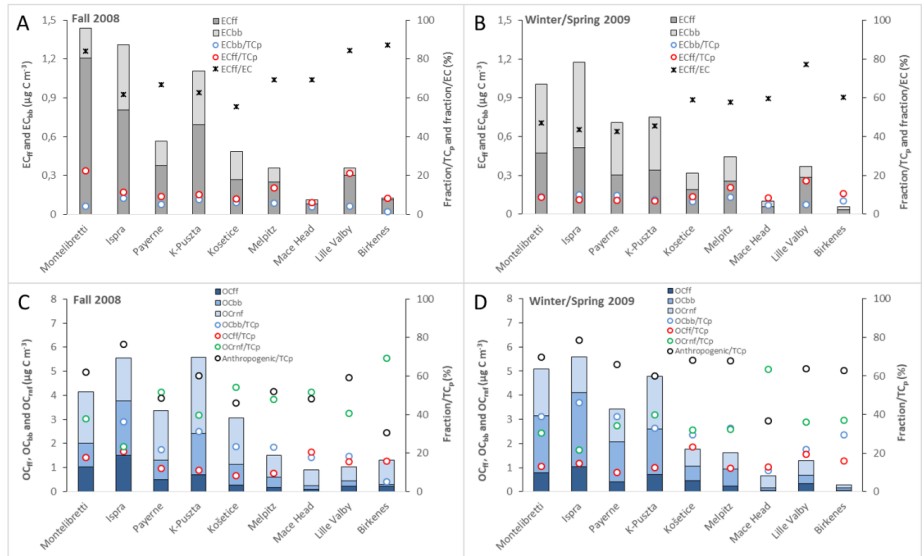

**Figure 2: Mass concentrations of EC from fossil fuel (ECff) and biomass burning (ECbb) sources, their fraction of particulate total carbon (TCp) and the fraction of ECff to EC for Fall 2008 (panel A) and Winter/Spring 2009 (panel B). Mass concentrations of OC from fossil fuel (OCff), biomass burning (OCbb) and remaining non-fossil (OCnrnf) sources, their fraction of TCp and the fraction of Anthropogenic (OCff, OCbb ECff and ECbb) to TCp for Fall 2008 (panel C) and winter/spring 2009 (panel D). The sites are listed by latitude from South to North. Note that the ECff/TCp marker is superimposed on the ECbb/TCp marker for Montelibretti and K-puszta in panel B, and that the OCff/TCp marker is superimposed on the OCbb/TCp marker for Montelibretti in panel C.**



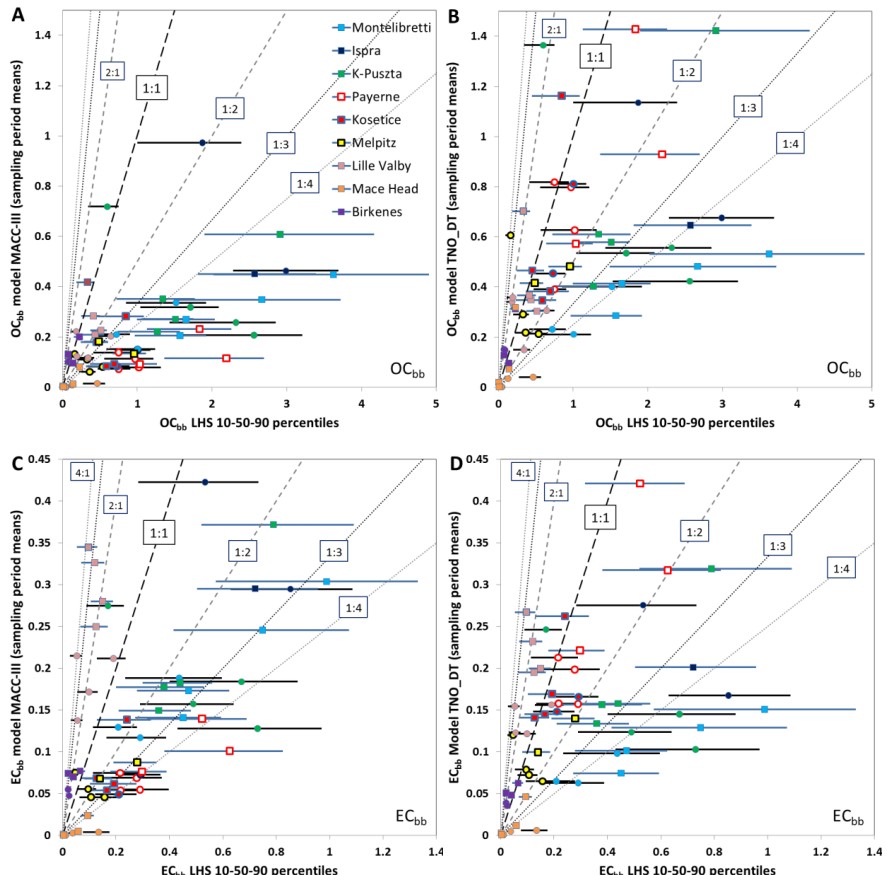

**Figure 3: Comparison of modelled and measurement/LHS based concentrations of organic and elemental carbon from biomass burning emissions (OC$_{bb}$ and EC$_{bb}$). The left panels (A and C) show model calculated OCbb (A) and ECbb (C) with the base-case model setup, and the right panels (B and D) show the corresponding results using the DT+IVOC model setup. Each point (and horizontal line) represents the results from a single site and week. The lines illustrate the range from the LHS 10-percentile to the 90-percentile and the circles and squares show the LHS-median values. Circles and black horizontal lines show results for Fall 2008 and squares and blue lines show results from Winter/spring 2009. The different sites are identified as follows: Light Blue – Montelibretti; Dark Blue – Ispra; Green – K-puszta; White with red border – Payerne; Red with blue border – Košetice; Yellow with black border – Melpitz; Pink – Lille Valby; Orange – Mace Head; Purple – Birkenes. Unit: µg C m$^{-3}$.**