# Peer review of "The EMEP Intensive Measurement Period campaign,"

_Atmospheric Chemistry and Physics, 2018_

## Referee Comment (RC1) · Anonymous Referee #1 · 6 Dec 2018

This paper describes how measurements from remote sites in Europe are used to estimate source apportionment of carbonaceous aerosols. While the overall methodology is sound, some additional discussion and clarification of the methodologies is needed before the paper is suitable for publication.

Major Comments:

Abstract: The abstract is very long, and hard to discern what is important versus what is less important. I suggest cutting the length by at least a third by focusing only on the most important findings. Much of the methodology can simply be left in the text and is not needed in the abstract.

[Figure]

Lines 163-220: The authors need to include some information on the size of aerosols that each of the instruments accounts for. I wonder if the size range is the same for each instrument. This is important when all the data sources are used together. If the size range is different, it will affect the source apportionment analysis. Not so much on the types of sources, but the relative contribution of the sources.

Long-Range Transport: There is no discussion in the text regarding the role of long-range transport of aerosols into Europe on the measurements that were collected and analyzed. The source apportionment assumes the source originate in Europe and the authors further speculate on uncertainties in European emissions inventories. It is possible that long-range transport will not be significant for certain time periods. The one-month sampling periods for the winter and spring period is relatively short, so the conclusions in this study may not be applicable over longer periods in general.

Modeling: The authors need to include some text on how the model accounts for long-range transport through its lateral boundaries (and how the initial conditions are generated and what type of spin up period is used). These results may or may not affect their analyses, depending how strongly the local emissions really explain the observed variability at the remote measurement sites. Some discussion on representativeness of the measurements is needed in the context of the 50 km grid spacing used. For some remote sites, the measurements may be representative over the 50 km grid. But this may not be the case for sites located in mountainous regions. The authors show the results of two emission scenarios, which will affect the amount of SOA produced by the model. What I would like to see is some additional discussion regarding how the model is used to speculate on errors in the emissions inventories. There are many SOA methodologies at present and one could get a range of answers in simulated organic matter.

Specific Comments:

Lines 92-93: The authors link carbonaceous aerosols to climate forcing and adverse

health effects; however, it seems to downplay the role of inorganic aerosol components on climate forcing and adverse health effects. For climate forcing and health effects, it is the total aerosol mass that matters. I understand the authors are trying to justify their work on studying carbonaceous aerosols (which often makes up a majority or large fraction of total aerosol mass), but the sentence they used is a bit misleading.

Line 302: Why is OC from biomass burning emissions treated as non-volatile? What makes that OC different from anthropogenic OC that is treated as volatile? Some discussion from the literature is needed to make this assumption, and my understanding is that whether biomass burning emissions are volatile and whether biomass burning emissions contributes significantly to SOA formation is still debatable.

Line 409: It is not clear what the plus/minus values mean. Are they the uncertainty range? Or are they a standard deviation? Please be specific.

Lines 438-440: The authors should try to explain why the EC/TCp ratio did not change much between the winter and spring period. I would have expected SOA to be more pronounced in the summer which would increase TCp. But maybe SOA formation is not that significant for those sites in the spring. Also, I am wondering what is the significance of the EC/TCp ratio? That is not described here, so it is difficult to know why readers should care about this ratio. The values are reported, but what is the significance?

Lines 518-519: The authors state that agricultural burning is banned, but I gather that it still happens. But if it was banned, why would it be a major source of air pollution? I think something is missing in the intent of this sentence which is confusing to me.

---

## Referee Comment (RC2) · Anonymous Referee #2 · 8 Dec 2018

This paper describes a quantification of the sources contributing to the organic carbon (OC) and elemental carbon (EC) components of samples of particulate matter (PM) collected at nine rural background locations spanning from Italy in the south to southern Norway in the north. PM samples were collected in two tranches: autumn (fall) 2008 and winter/spring 2009.

A standard approach to the source apportionment has been applied. Chemically, the samples have been spit into OC and EC concentrations via the EUSAAR-2 temperature programme, analysed for the carbon-14 content of the total carbon (TC) using accelerator mass spectrometry, and quantified for levoglucosan, mannosan and galactosan

tracers via HPLC-MS. These measurements are used to provide apportionment of the TC into a set of natural vs anthropogenic and primary vs secondary sources. To assess the impact of uncertainties in the apportionment, a sensitivity analysis on the apportionment is performed using a Latin hypercube approach to sampling within uncertainty ranges on apportionment parameters. This methodology has been applied a number of times before by combinations of these authors in similar carbonaceous aerosol source apportionment studies. It is supplemented in this work with use of atmospheric chemistry transport modelling to both help refine apportionment, e.g. to estimate the proportion of wildfire and agricultural burning within the overarching biomass-burning OC source, and to evaluate the closure of model simulated OC and OC with measured OC and EC. The latter reveals that biomass burning sources of carbonaceous aerosols are underestimated in standard national emissions inventories.

The analysis is comprehensive and the presentation of the work in the paper is likewise comprehensive, clear and accurate.

A question is how relevant do the results remain for present day, given that the samples were collected 10 years ago? Other than that I find the paper to be suitable for publication with only very few typographical corrections, and ACP is an appropriate journal for this work.

I was pleased to read the honesty of the authors about some PM samples having unfeasibly large carbon-14 values, the origin of which cannot be pinpointed. Whilst it is easy to identify samples with carbon-14 values that are way out of line with what can be expected for 'normal' samples, there is the concern that other carbon-14 values that are not way out of line may also be unknowingly 'contaminated' in some way and thus yield error in apportionment that is not recognised. At present, there seems no way to resolve this.

Minor formatting errors:

L472: should refer to sections 4.1-4.5.

L476: delete the duplicate "same".

L509: should refer to section 4.2.

L597: replace "barely" with "only".
* * *

---

## Author Response (AR1)

**Reply to referee #1:**

**Major comments:**

**Abstract**: The abstract is very long, and hard to discern what is important versus what is less important. I suggest cutting the length by at least a third by focusing only on the most important findings. Much of the methodology can simply be left in the text and is not needed in the abstract.

**Answer:**

The abstract was shortened from 527 words to 369 words (corresponding to 30%), and along the guidelines suggested by the referee.

**Action:**

**Revised abstract has been included in the revised paper.**

Carbonaceous aerosol (Total Carbon;  $TC_p$ ) was source apportioned at nine European rural background sites, as part of the EMEP Intensive Measurement Periods in fall 2008 and winter/spring 2009. Five predefined fractions were apportioned based on ambient measurements: Elemental and organic carbon from combustion of biomass ( $EC_{bb}$  and  $OC_{bb}$ ) and from fossil fuel ( $EC_{ff}$  and  $OC_{ff}$ ) sources, and remaining non-fossil organic carbon ( $OC_{rnf}$ ), dominated by natural sources.

 $OC_{rnf}$  made a larger contribution to  $TC_p$  than anthropogenic sources (ECbb,  $OC_{bb}$ , ECff and  $OC_{ff}$ ) at four out of nine sites in fall, reflecting the vegetative season, whereas anthropogenic sources dominated at all but one site in winter/spring. Biomass burning ( $OC_{bb}+EC_{bb}$ ) was the major anthropogenic source at the Central European sites in fall, whereas fossil fuel ( $OC_{ff}+EC_{ff}$ ) sources dominated at the southernmost and the two northernmost sites. Residential wood burning emissions explained 30-50% of  $TC_p$  at most sites in the first week of sampling in fall, showing that this source can be dominating even outside the heating season. In winter/spring, biomass burning was the major anthropogenic source at all but two sites, reflecting increased residential wood burning emissions in the heating season. Fossil fuel sources dominated EC at all sites in fall, whereas there was as shift towards biomass burning for the southernmost sites in winter/spring.

Model calculations based on base-case emissions (mainly officially reported national emissions) strongly underpredicted observational derived levels of  $OC_{bb}$  and  $EC_{bb}$  outside Scandinavia. Emissions based on a consistent bottom-up inventory for residential wood burning (and including intermediate volatility compounds, IVOC), improved model results compared to the base-case emissions, but modelled levels were still substantially underestimated compared to observational derived  $OC_{bb}$  and  $EC_{bb}$  levels at the southernmost sites.

Our study shows that natural sources is a major contributor to carbonaceous aerosol in Europe even in fall and in winter/spring, and that residential wood burning emissions are equally large or larger than that of fossil fuel sources, depending on season and region. The poorly constrained residential wood burning emissions for large parts of Europe shows the obvious need to improve emission inventories, with harmonization of emission factors between countries likely being the most important step to improve model calculations for biomass burning emissions, and European PM2.5 concentrations in general.

**Lines 163-220**: The authors need to include some information on the size of aerosols that each of the instruments accounts for. I wonder if the size range is the same for each instrument. This is important when all the data sources are used together. If the size range is different, it will affect the source apportionment analysis. Not so much on the types of sources, but the relative contribution of the sources.

**Answer:**

We understand the question as if the referee is asking whether the aerosol filter sampler operated at each of the nine sites had the same aerosol particle cut-off size or not, and that information about the samplers aerosol particle cut-off size should be provided in the text.

Indeed, this information is already present in the paper. Lines 155 – 157 states that:

"Ambient aerosol filter samples were obtained using various low volume filter samplers equipped with a PM10 inlet, collecting aerosol on prefired (850 °C; 3 h) quartz fiber filters (Whatman QMA; 47 mm in diameter, batch number 11415138). The only exception was for samples collected at the Mace Head station, which used a high volume sampler with a PM2.5 inlet."

Hence, all sites used a filter sampler with a PM10 inlet, except for Mace Head, which had a sampler with a PM2.5 inlet. We also sampled according to the QBQ approach, picked quartz fibre filters from the same batch number, and used centralized laboratories to minimize differences between sites that are not related to the samplers' cut-off size. Information about the cut-off size for each of the sites can also be found in Table 1, 2a, and 2b.

The most likely effect of aerosol filter samples collected with a PM2.5 inlet instead of a PM10 inlet at Mace Head, is an underestimation of OCrnf, as most primary biological aerosol particles reside in the coarse fraction of PM10. Combustion of fossil fuel and biomass generates OC and EC in the fine fraction of PM10, consequently the influence of a PM2.5 cut-off size is minor for these. A higher OCrnf level resulting from a PM10 inlet would not change any of our conclusions, as OCrnf was the highest fraction at Mace Head regardless of season, even when based on a PM2.5 inlet.

**Action:**

We have included the following two sentences to underpin that the effect of PM2.5 aerosol filter samples collected at Mace Head do not change any of our conclusions in section **4.3**, **lines 606 – 610**.

Note that  $OC_{rnf}$  obtained for Mace Head is a conservative estimate, as PBAP typically residing in the coarse fraction is not accounted for, as  $PM_{2.5}$  filter samples were collected at this site. Nevertheless,  $OC_{rnf}$  was the major fraction at Mace Head, regardless of season; hence, our conclusions would not change if the filter samples had  $PM_{10}$  cut-off size.

**Long-Range Transport**: There is no discussion in the text regarding the role of long-range transport of aerosols into Europe on the measurements that were collected and analyzed. The source apportionment assumes the source originate in Europe and the authors further speculate on uncertainties in European emissions inventories. It is possible that long-range transport will not be significant for certain time periods. The one-month sampling periods for the winter and spring period is relatively short, so the conclusions in this study may not be applicable over longer periods in general.

**Answer:**

The issue of long-range transport into Europe is important for some pollutants (especially ozone, e.g. Fiore et al., 2009, or carbon monoxide from forest fires, e.g. Forster et al., 2001). However, many years of measurements and modelling analyses support our assumption that the most likely sources of carbonaceous aerosols in our study are from Europe. For example, many years of analysis of aerosols at Mace Head on the west coast of Ireland give

little evidence for aerosol transport from North America, with most organic matter (OM) assigned to marine or European sources (O'Dowd et al., 2014). Emissions from major wildfires in Eastern Europe explained the highest OC and EC concentrations at Birkenes in 2001 – 2015, as did episodes of air pollution carrying the hallmark of long-range transport; i.e., elevated levels of secondary inorganic aerosol and air masses transported at low altitude over major emission regions in Central and Eastern Europe (Yttri et al. in prep.). Meanwhile, elevated concentrations of equivalent black carbon (eBC) from fossil fuel sources (eBCff) and from biomass burning (eBCff) at Birkenes were associated exclusively with source regions in continental Europe (Yttri et al., in prep). Consequently, long-range transport is of major importance for elevated concentrations of carbonaceous aerosol at Birkenes, but sources are confined to the European continent.

Further, modelling by Simpson et al. (2007) showed that observed levels of OC and EC could be reproduced quite well over a 2-year period (CARBOSOL study) at two sites on the western coast of Europe, Mace Head in Ireland, and Aveiro in Portugal, with no suggestion of missing background sources in the model. Tsyro et al. (2007) examined the EC concentrations for the same study, and showed that European forest fires only had significant impacts for a few samples. We note that the modelling domain we use is rather large, covering all of Europe from approximately 40 degree W to 60 degree E and 30-90 degree N, such that we capture all major sources and air mass circulations within several days of transport. Global model results from the EMEP model (e.g. McFiggans et al., 2019) also suggest that OM generated over North America makes only a small contribution to European particulate matter levels.

The referee comments that the conclusions of our study are for a short period and may thus not be generally applicable. If the current study was the only one, this would be true. However, there are multiple studies that have pointed out the problems with European residential wood burning inventories for both OM and EC (Simpson et al., 2007, Genberg et al., 2011, 2013, Bergström et al., 2012, Denier van der Gon et al., 2015a), and the conclusions of the current study reinforce these earlier results.

**Action:**

*We have added the two first paragraphs of our answer to the referee to the Discussion part of the manuscript.* (4.6. *Influence of long range transport*).

We have added the following sentence to the **Conclusion (lines 761 -765)** part of the manuscript to answer the referees comment.

Although the results of this particular study are for two relatively short periods, the general conclusions are consistent with those from multiple studies, which have pointed out the problems with European RWC inventories for both OC and EC (Simpson et al., 2007, Genberg et al., 2011, 2013, Bergström et al., 2012, Denier van der Gon, 2015a). The conclusions of the current study complement and reinforce these earlier results.

**Modeling**: The authors need to include some text on how the model accounts for long-range transport through its lateral boundaries (and how the initial conditions are generated and what type of spin up period is used). These results may or may not affect their analyses, depending how strongly the local emissions really explain the observed variability at the remote measurement sites. Some discussion on representativeness of the measurements is needed in the context of the 50 km grid spacing used. For some remote sites, the measurements may be representative over the 50 km grid. But this may not be the case for sites located in mountainous regions. The authors show the results of two emission scenarios, which will affect the amount of SOA produced by the model. What I would like to see is some additional discussion regarding how the model is used to speculate on errors in the emissions inventories. There are many SOA methodologies at present and one could get a range of answers in simulated organic matter.

**Answer:**

We have added text on the model implementation of boundary conditions of OM as detailed below (see "Action"). The general issue of long-range transport and boundary conditions has been addressed above. With regard to

spin-up, the model simulations performed in this study covered the full years 2008 and 2009, which means that the spin up period for the Winter/spring period was 54 days and more than eight months for the Fall period.

Concerning the question of site representativity, we agree that this is a challenging topic in general, and not only for mountainous areas with substantial vertical variability, as mentioned by the referee. An assessment of the representativity of European air quality measurement sites is very well described by Henne et al. (2010) (also referred to in the paper), including five of the nine sites included in the present study, whereas a general description of the measurement sites and their surroundings is given in Appendix A of the present paper. In general, EMEP sites are specifically chosen to be as representative as possible for these larger grid squares.

The referee is correct in stating that different SOA schemes give different answers, as we explored in detail in Bergström et al. (2012). However, sensitivity tests performed as part of the studies by Bergström et al. (2012), Simpson et al. (2012) and Denier van der Gon et al. (2015a) have shown that differences in OM caused by emissions assumptions are larger than those caused by e.g. volatility assumptions. We have used two sets of assumptions (base-case and DT+IVOC), which we believe span a reasonable range of possibilities. However, to answer the referee without adding too much text to the manuscript we have added a small section on the initial/boundary conditions of the carbonaceous aerosol to **section 1.7**, and on model uncertainties in the **Discussion** part of the Manuscript (see "Actions").

**Action:**

Concerning the initial/boundary conditions for the carbonaceous aerosol, we have added the following text in the end of **section 1.7, lines 280-287**:

Initial and lateral boundary conditions for the EMEP model are specified for most pollutants, as in Simpson et al. (2012). For OM, the model assumes a background level of organic matter to represent OM transported into the modelling domain, or otherwise not accounted for (e.g. marine aerosol, some primary biological aerosol particles, or very aged aerosol from outside the domain). In the initial setup of Bergström et al. (2012) and Simpson et al. (2012), we used 1.0 ug m-3 OM, but results presented in Bergström et al. (2012) and later studies suggested that this was too high. As in Bergström et al. (2014), we assume a background concentration of particulate OM of 0.4 ug m-3 (with an OM/OC ratio of 2.0) near the ground.

**Concerning SOA schemes, we have added the following new paragraph to the end of section **4** in the **Discussion**, **lines 481-497**:**

A major difficulty for all modelling work is the complexity of organic aerosol, in terms of emissions, formation mechanisms, and deposition processes (e.g. Hallquist et al., 2009; Hodzic et al., 2016). Considering emissions, we can note that Denier van der Gon (2015a) utilized a specially developed map of residential wood combustion sources, which however was specific to that study and not utilized in subsequent spatial mapping of emissions. Studies in the United Kingdom and Norway have also cast doubt on the accuracy of spatial distributions of emissions (Ots et al., 2016; López-Aparicio et al., 2017), which inevitably causes problems for modelling. Compounding the difficulties, different SOA schemes give different answers, as we explored in detail in Bergström et al. (2012). However, sensitivity tests performed as part of the studies by Bergström et al. (2012), Simpson et al. (2012) and Denier van der Gon et al. (2015a) have shown that differences in OM caused by emissions (base-case and DT+IVOC) in our work, which we believe span a reasonable range of possibilities. Given these difficulties, it is not surprising that model results can show large scatter compared to measured values. However, we have also shown in several studies (Bergström et al., 2012, Genberg et al., 2011, 2013, Denier van der Gon et al., 2012, Genberg et al., 2011, 2013, Denier van der Gon et al., 2015a), that the model results do improve compared to observations when condensables are treated in a more uniform matter, and the current study is consistent with this.

**Specific Comments:**

**Lines 92-93:** The authors link carbonaceous aerosols to climate forcing and adverse health effects; however, it seems to downplay the role of inorganic aerosol components on climate forcing and adverse health effects. For climate forcing and health effects, it is the total aerosol mass that matters. I understand the authors are trying to justify their work on studying carbonaceous aerosols (which often makes up a majority or large fraction of total aerosol mass), but the sentence they used is a bit misleading.

**Answer:**

Our intension is not to downplay the importance of inorganics in relation to climate forcing and adverse health effects of the atmospheric aerosol. Rather the paper is about the carbonaceous fraction of the aerosol, thus we aim to narrow the text accordingly, not to make it too extensive.

We have rewritten the sentence slightly to indicate that the carbonaceous aerosol is not the only aerosol fraction of the atmospheric aerosol contributing to climate forcing and adverse health effects.

**Action:**

We have rewritten the original sentence slightly and included the following sentence (lines 83 – 85):

Because of its influence on climate forcing and adverse health effects, as well as its considerable contribution to particulate mass, source apportionment of carbonaceous aerosol is of key importance.

**Line 302:** Why is OC from biomass burning emissions treated as non-volatile? What makes that OC different from anthropogenic OC that is treated as volatile? Some discussion from the literature is needed to make this assumption, and my understanding is that whether biomass burning emissions are volatile and whether biomass burning emissions contributes significantly to SOA formation is still debatable.

**Answer:**

This was a simplification, so we had a clear tracer of these emissions, but where we could concentrate on the modelling of the residential biomass burning emissions. Primary organic aerosol emissions from open burning of biomass (wildfires and agricultural burning etc.) is also a mixture of gaseous and particle components, spanning the VBS system (e.g. May et al., 2013). Indeed, Bergström et al. (2012) implemented such a system, and found that there is a relatively large potential for SOA formation from IVOC emitted from biomass burning during summer. However, given the relatively small contribution of this source during the time periods covered in this study, and the very large uncertainties regarding the volatility distribution, and especially aging of the primary organic aerosol and IVOC emissions from biomass burning, we preferred to adopt the simple assumption of inert primary organic aerosol compounds, in order to have a tracer species for these sources, which is easy to interpret.

**Action:**

In order to keep the text on the modelling setup reasonably concise, we have simply modified the sentence in Sect. **1.7, line 272-274**, from 'for simplicity' to 'in order to provide a tracer of these emissions, but without adding the considerable uncertainties associated with aging of any assumed VBS components."

Line 409: It is not clear what the plus/minus values mean. Are they the uncertainty range? Or are they a standard deviation? Please be specific.

**Answer:**

The  $\pm$  indicates the standard deviation. We have clarified this in the text by adding ( $\pm$  SD; standard deviation) not only for line 409 but also for line 410, 425, and 428.

**Action:**

The  $\pm$  indicates the standard deviation and we have clarified this in the text by adding ( $\pm$  SD; standard deviation) not only for line 409 but also for line 410, 425, and 428 and in Table 2a and Table 2b.

**Lines 438-440**: The authors should try to explain why the  $EC/TC_p$  ratio did not change much between the winter and spring period. I would have expected SOA to be more pronounced in the summer, which would increase  $TC_p$ . But maybe SOA formation is not that significant for those sites in the spring. Also, I am wondering what is the significance of the  $EC/TC_p$  ratio? That is not described here, so it is difficult to know why readers should care about this ratio. The values are reported, but what is the significance?

**Answer:**

*We have extracted the following to questions from the referees comment:*

"What is the significance of the EC/TCp ratio?" and "the authors should try to explain why the EC/TCp ratio did not change much between the winter and spring period."

Note: The two periods studied in the present paper is Winter/spring and Fall, not winter and spring as can be understood from the referees question.

Albeit crude, EC is a tracer of anthropogenic activity, thus the EC/TCP ratio indicates combustion derived anthropogenic primary aerosol particles' influence of the carbonaceous aerosol, and thus should accompany reported levels of EC. Except from being such an indicator, the EC/TCP ratio does not serve any purpose in the present study.

The referee is correct in stating that the  $EC/TC_P$  ratio is not used to interpret the data in any broader sense in the present study. The reason for this is that it is not useful to apply the  $EC/TC_P$  ratio to speculate about source contribution in a study that was designed to do source apportionment based on a much more sophisticated approach (here: 14C-TC, levoglucosan, OC, and EC analysis in combination with appropriate emission factors treated statistically using Latin Hypercube sampling), and which separates EC into biomass burning and fossil fuel origin. The  $EC/TC_P$  ratio observed in the two measurement periods is a result of the contributing sources (which have been apportioned) and their relative share as a function of season is thoroughly discussed in the present study, but not to explain the  $EC/TC_P$  ratio.

**Action:**

No action needed.

**Lines 518-519**: The authors state that agricultural burning is banned, but I gather that it still happens. But if it was banned, why would it be a major source of air pollution? I think something is missing in the intent of this sentence which is confusing to me.

**Answer:**

The answer to your question comes in the sentence following that of lines 518-519, i.e. **lines 537-540** (see underlined text):

Agricultural waste burning is banned in most European countries, as it is a major source of forest fires, and thus a threat to human life and properties, as well as a source of severe air pollution. Nevertheless, remote sensing data show such fire events in several countries, including those with a ban (Korontzi et al., 2006), and it appears particularly frequent in Eastern Europe (e.g. Belarus and the Ukraine), in western parts of Russia and in Central Asia.

The text on **lines 537-540** states that despite the ban, such fires occur, and can contribute. We have cut a part of the first sentence, and merged the two sentences to improve the readability.

**Action:**

We have rewritten the original sentence slightly and included the following sentence (line 537-540):

Agricultural waste burning is banned in most European countries, nevertheless, remote sensing data show such fire events in several countries, including those with a ban (Korontzi et al., 2006), and it appears particularly frequent in Eastern Europe (e.g. Belarus and the Ukraine), in western parts of Russia, and in Central Asia.

**Reply to referee #2:**

**General comment:**

A question is how relevant do the results remain for present day, given that the samples were collected 10 years ago? Other than that, I find the paper to be suitable for publication with only very few typographical corrections, and ACP is an appropriate journal for this work.

**Answer:**

Despite that the filter samples being the basis for the reported carbonaceous aerosol source apportionment study was collected ten years ago, they still provide useful information documenting the carbonaceous aerosol sources in Europe. Starting with the paper of Gelencsér et al. (2007), there are very few measurement-based studies source apportioning carbonaceous aerosol in Europe on a larger scale. Hence, the current study fill in gaps in time and space, as well as by methodology. As to what extent the results are relevant for the present day, papers that are more recent in time point to the same pattern as reported in the present study; increased emissions from residential wood burning in winter, fossil fuel sources typically dominating EC, and natural sources being a major source in the vegetative season.

**Action:**

No action needed.

**Minor formatting errors:**

L474: should refer to sections 4.1-4.6.

Action:

"5.1 – 5.5" replaced by "4.1 – 4.6"

L478: delete the duplicate "same".

**Action:**

"same same" has been replaced by "same"

L509: should refer to section 4.2.

**Action:**

L528: "5.2" replaced by "4.2"

L597: replace "barely" with "only".

**Action:**

L619: "barely" replaced by "only"

**Action:**

We have made the following changes and additions to the paper separate from the comments made by Referee #1 and #2:

Sect 1.7, L278 (earlier L286)- should refer to Sects. 1.7.1.1 and 1.7.1.2, not 2.7...

Sect 4.4, Page 17, L637 (earlier L615). Included the following reference: Andersson-Sköld, Y. & Simpson, D., Secondary organic aerosol formation in Northern Europe: a model study, J. Geophys. Res., 2001, 106, 7357-7374.

Sect. 4.5, L657 (earlier L35) Should be Sect. 1.7, not section 2.7.

New references has been added to the paper, and to the reference list, as part of the reviewing process

**References used in our reply to referee #1 and referee #2.**

[revised manuscript text omitted]

Yttri et al. (In prep.)

**1 The EMEP Intensive Measurement Period campaign,**

**2 2008–2009: Characterizing the carbonaceous aerosol at**

**3 nine rural sites in Europe**

4 Karl Espen Yttri\*a, David Simpsonb,c, Robert Bergströmc,d, Gyula Kisse, Sönke
5 Szidatf, Darius Ceburnisg, Sabine Eckhardta, Christoph Hueglinh, Jacob Klenø

5 beldut, buttus coournes, subme lexinater, emistoph filogini, succo kieno

- 6 Nøjgaardi, Cinzia Perrinoj, Ignazio Pissoa, Andre Stephan Henry Prevotk, Jean-
- 7 Philippe Putaud1, Gerald Spindlerm, Milan Vanan, Yan-Lin Zhangjkm, Wenche Aasa
- 8

9 aNILU — Norwegian Institute for Air Research (NILU), N-2027 Kjeller, Norway

- 10 bNorwegian Meteorological Institute, 0313 Oslo, Norway
- 11 cDepartment of Space, Earth and Environment, Chalmers University of Technology, 41296 Gothenburg
- 12 dSwedish Meteorological and Hydrological Institute, 60176 Norrköping, Sweden
- 13 °MTA-PE Air Chemistry Research Group, 8200 Veszprém Hungary
- 14 fDepartment of Chemistry and Biochemistry & Oeschger Centre for Climate Change Research,
- 15 University of Bern, 3012 Berne, Switzerland
- 16 gSchool of Physics and Centre for Climate and Air Pollution Studies, Ryan Institute, National
- 17 University of Ireland Galway, Galway, Ireland
- 18 hEMPA, CH-8600 Duebendorf, Switzerland
- 19 iNational Environmental Research Institute, DK-4000 Roskilde, Denmark
- 20 jCNR Institute of Atmospheric Pollution Research, 00015 Monterotondo Stazione (Rome), Italy
- 21 kPaul Scherrer Institut, 5232 Villigen-PSI, Switzerland
- 22 IEuropean Commission, Joint Research Centre, I-21027 Ispra (VA), Italy
- 23 mLeibniz Institute for Tropospheric Research , 04318 Leipzig, Germany
- 24 nThe Czech Hydrometeorological Institute (CHMI), Prague, Czech Republic

- 26 \*To whom correspondence should be addressed: E-mail address: key@nilu.no
- 27

28 Abstract

Carbonaceous aerosol (Total Carbon;  $TC_p$ ) was source apportioned at nine European rural background sites, as part of the EMEP Intensive Measurement Periods in fall 2008 and winter/spring 2009. Five predefined fractions were apportioned based on ambient measurements: Elemental and organic carbon from combustion of biomass (ECbb and OCbb) and from fossil fuel (ECff and OCff) sources, and remaining non-fossil organic carbon (OCrnf), 
[revised manuscript text omitted]

| 1090 | (100CAARC) - integrating acrossil research from nano to grobal scales, Atmos. Chem. 1 hys., 7, 2025-
2841. doi:10.5104/acp.0.2825.2000.2000   |
| 1001 | 2041, doi.10.5194/acp-9-2025-2009, 2009.                                                                                                         |
| 1092 | Liu I Li I Vonwiller M Liu D. Cheng H. Shen K. Selazar G. Agrios K. Zhang V. Hea O.                                                              |
| 1003 | Ding V. Zhong C. Wong V. Szidet S. and Zhong C. The importance of non-facel courses in                                                           |

[revised manuscript text omitted]
                     | $6.1\pm2.7$   | $9.3\pm5.7$   | $3.6\pm1.3$   | $5.5\pm2.8$   | $2.1\pm0.78$  | $1.7\pm0.68$  | $0.76\pm0.91$          | $1.5\pm0.33$  | $0.44\pm0.13$ |
| $OC_p$                              | $5.0\pm2.5$   | $7.9\pm5.0$   | $2.9 \pm 1.0$ | $4.8\pm2.6$   | $1.8\pm0.70$  | $1.3\pm0.50$  | $0.65\pm0.79$          | $1.2\pm0.3$   | $0.34\pm0.08$ |
| OC Back                  | $0.62\pm0.16$ | $0.50\pm0.22$ | $0.41\pm0.18$ | $0.35\pm0.10$ | $0.23\pm0.09$ | $0.41\pm0.26$ | $0.07\pm0.04$          | $0.53\pm0.31$ | $0.13\pm0.13$ |
| EC                                  | $1.0\pm0.25$  | $1.5\pm0.68$  | $0.66\pm0.27$ | $0.77\pm0.21$ | $0.32\pm0.12$ | $0.40\pm0.12$ | $0.11\pm0.13$          | $0.37\pm0.09$ | $0.10\pm0.05$ |
| Unit: (%)                           |               |               |               |               |               |               |                        |               |               |
| EC/TC p                  | $18\pm3.6$    | $17\pm2.3$    | $19\pm2.9$    | $15 \pm 3.3$  | $16\pm1.4$    | $24 \pm 4.1$  | $14\pm1.3$             | $24\pm5.4$    | $21\pm5.2$    |
| $OC_{Back} / OC_{Front}$            | $12 \pm 2.9$  | $6.6\pm1.3$   | $12 \pm 1.9$  | $7.3 \pm 1.4$ | $12 \pm 4.4$  | $24 \pm 12$   | $23 \pm 21$            | $30 \pm 10$   | $24 \pm 13$   |
| Unit: (Fraction)                    |               |               |               |               |               |               |                        |               |               |
| f nf (TC p )  | $0.80\pm0.06$ | $0.80\pm0.05$ | $0.90\pm0.09$ | $0.83\pm0.09$ | $0.69\pm0.04$ | $0.83\pm0.13$ | $0.79\pm0.11$          | $0.71\pm0.13$ | $0.77\pm0.09$ |
| Unit: (ng m-3)    |               |               |               |               |               |               |                        |               |               |
| Levoglucosan                        | 247 ± 113     | $668\pm295$   | $141 \pm 63$  | $209 \pm 156$ | 67 ± 16       | $57 \pm 20$   | $12 \pm 13$            | $41 \pm 5.5$  | $17 \pm 7.7$  |

Table 2a: Mean (± SD; standard deviation) concentrations of carbonaceous sub-fractions and levoglucosan in  $PM_{10}^1$  during Winter/Spring 2009. The EC/TCp ratio, the OCBack/OCFront ratio and non-fossil fractions of TCp (fnf(TCp)) are also listed. The sites are ordered by latitude from south to north.

1) For Mace Head PM2.5 was used

|                                            | Montelibretti 2 | Ispra         | Payerne       | K-puszta      | Košetice      | Melpitz       | Mace Head 1 | Lille Valby   | Birkenes        |
|--------------------------------------------|----------------------------|---------------|---------------|---------------|---------------|---------------|------------------------|---------------|-----------------|
| Unit: $(\mu g \ C \ m^{-3})$        |                            |               |               |               |               |               |                        |               |                 |
| TCp                                        | $5.0 \pm 1.8$              | $7.6\pm2.5$   | $3.9 \pm 1.1$ | $6.7\pm2.9$   | $3.3\pm0.66$  | $2.1\pm0.36$  | $0.89 \pm 1.2$         | $1.8\pm0.74$  | $1.1\pm0.47$    |
| $OC_p$                                     | $4.0 \pm 1.8$              | $6.1\pm2.0$   | $3.3\pm0.93$  | $5.5\pm2.7$   | $2.8\pm0.59$  | $1.6\pm0.21$  | $0.77 \pm 1.1$         | $1.3\pm0.70$  | $0.97 \pm 0.45$ |
| OC Back                         | $0.75\pm0.16$              | $0.47\pm0.31$ | $0.53\pm0.37$ | $0.33\pm0.08$ | $0.21\pm0.08$ | $0.60\pm0.33$ | $0.10\pm0.07$          | $0.48\pm0.21$ | $0.17\pm0.03$   |
| EC                                         | $0.97\pm0.25$              | $1.5\pm0.54$  | $0.59\pm0.17$ | $1.2\pm0.26$  | $0.49\pm0.10$ | $0.54\pm0.16$ | $0.12\pm0.17$          | $0.46\pm0.10$ | $0.11\pm0.03$   |
| Unit: (%)                                  |                            |               |               |               |               |               |                        |               |                 |
| EC/TC p                         | $21\pm8.3$                 | $20\pm3.7$    | $15\pm0.31$   | $18\pm4.0$    | $15 \pm 2.1$  | $25 \pm 3.7$  | $12\pm5.6$             | $28\pm8.1$    | $11 \pm 3.3$    |
| $OC_{Back} / OC_{Front}$                   | $17 \pm 3.8$               | $6.8\pm2.6$   | $13 \pm 4.9$  | $5.9\pm1.0$   | $6.9\pm1.5$   | $26\pm10$     | $19\pm8.9$             | $28 \pm 13$   | $19\pm6.7$      |
| Unit: (Fraction)                           |                            |               |               |               |               |               |                        |               |                 |
| f nf (TC p )         | $0.61\pm0.01$              | $0.69\pm0.08$ | $0.80\pm0.06$ | $0.81\pm0.03$ | $0.86\pm0.10$ | $0.76\pm0.04$ | $0.70\pm0.18$          | $0.72\pm0.12$ | $0.75\pm0.05$   |
| Unit: ( ng m -3 ) |                            |               |               |               |               |               |                        |               |                 |
| Levoglucosan                               | $106\pm40$                 | $364 \pm 180$ | $85\pm16$     | $172\pm84$    | $83 \pm 14$   | $33 \pm 14$   | $16 \pm 19$            | $32\pm19$     | $6.8\pm2.2$     |

Table 2b: Mean  $(\pm SD; standard deviation)$  concentrations of carbonaceous sub-fractions and levoglucosan in PM101 during Fall 2008. The EC/TCp ratio, the OCBack/OCFront ratio and non-fossil fractions of TCp (fnf(TCp)) are also listed. The sites are ordered from by latitude south to north.

1) For Mace Head PM2.5 was used.

2) The sampler at Montelibretti was run in an alternating on/off mode, collecting ambient air 15 minutes every 1 hour.

| $C^{*} (\mu g \ m^{-3})^{a}$ |                   | 10 -2 | 10 -1 | 1     | 10    | 10 2 | 10 3 | 10 4 | 10 5 | 10 6 |
|------------------------------|-------------------|------------------|------------------|-------|-------|------------------------|------------------------|------------------------|------------------------|------------------------|
| Base-case                    | SNAP 2            | 0.20             | 0.00             | 0.10  | 0.10  | 0.20                   | 0.40                   | 0.00                   | 0.00                   | 0.00                   |
| emission                     | all other sources | 0.00             | 0.04             | 0.25  | 0.37  | 0.23                   | 0.11                   | 0.00                   | 0.00                   | 0.00                   |
| fraction b        |                   |                  |                  |       |       |                        |                        |                        |                        |                        |
|                              |                   |                  |                  |       |       |                        |                        |                        |                        |                        |
| DT+IVOC                      | SNAP 2            | 0.025            | 0.050            | 0.076 | 0.118 | 0.151                  | 0.252                  | 0.336                  | 0.42                   | 0.672                  |
| emission                     | all other sources | 0.03             | 0.06             | 0.09  | 0.14  | 0.18                   | 0.30                   | 0.40                   | 0.50                   | 0.80                   |
| fraction c, d     |                   |                  |                  |       |       |                        |                        |                        |                        |                        |

Table 3: Volatility distributions of the primary organic aerosol (POA) emissions from anthropogenic sources.

a C\*: Saturation concentration at 298 K; enthalpies of vaporization were taken from May et al. (2013a,b) for the base-case (MACC-III), and from Shrivastava et al. (2008) for the DT+IVOC case.

b The volatility distribution in the MACC-III model run is based on the recommended volatility distributions from May et al. (2013a,b) for biomass burning emissions (for SNAP sector 2; non-industrial stationary combustion) and for diesel exhaust (for all the other emission sectors), but moving the emissions in the  $C^*=10^4 \ \mu g \ m^{-3}-10^6 \ \mu g \ m^{-3}$  bins to the  $10^3 \ \mu g \ m^{-3}$  bin.

c The volatility distributions in the DT+IVOC case are based on Shrivastava et al. (2008) for all emission sectors except SNAP-2, for which it is based on the distribution used for the EMEP model in Denier van der Gon et al. (2015a). Note that this scenario assumes that there are substantial IVOC emissions that are not included in the emission inventories (see Bergström et al., 2012, and Denier van der Gon et al., 2015a).

d Since the DT emission inventory by Denier van der Gon et al. (2015a) was constructed to include a larger fraction of SVOC from residential wood burning emissions, we apply a slightly different emission split for the SNAP-2 POA compared to other SNAP sectors. Considering both SVOC and IVOC within the POA class, the total POA emissions are assumed to be 2.1 times the inventory (compared to the factor 2.5 for the other emission sectors).

|               |           | Ε       |        | OC bb |           |         |        |        |
|---------------|-----------|---------|--------|------------------|-----------|---------|--------|--------|
| Site          | Base-case | DT+IVOC | LHS-10 | LHS-90           | Base-case | DT+IVOC | LHS-10 | LHS-90 |
| Montelibretti | 0.19      | 0.097   | 0.29   | 0.70             | 0.28      | 0.37    | 1.04   | 2.38   |
| Ispra         | 0.34      | 0.21    | 0.47   | 0.93             | 0.63      | 0.82    | 1.70   | 3.16   |
| K-puszta      | 0.20      | 0.17    | 0.30   | 0.67             | 0.37      | 0.74    | 1.10   | 2.27   |
| Payerne       | 0.081     | 0.24    | 0.20   | 0.46             | 0.12      | 0.79    | 0.73   | 1.51   |
| Košetice      | 0.074     | 0.17    | 0.12   | 0.28             | 0.14      | 0.60    | 0.42   | 0.91   |
| Melpitz       | 0.063     | 0.096   | 0.085  | 0.18             | 0.12      | 0.37    | 0.30   | 0.57   |
| Mace Head     | 0.0045    | 0.0091  | 0.028  | 0.057            | 0.015     | 0.061   | 0.086  | 0.16   |
| Lille Valby   | 0.24      | 0.18    | 0.067  | 0.14             | 0.22      | 0.36    | 0.24   | 0.46   |
| Birkenes      | 0.065     | 0.047   | 0.020  | 0.046            | 0.13      | 0.17    | 0.072  | 0.15   |

Table 4: Model and source apportioned (LHS-derived) concentrations of elemental carbon (ECbb) and organic carbon (OCbb) from biomass burning. Model results are averages over both measurement periods (Fall 2008 and Winter/Spring 2009). For the LHS-results the mean of the 10- and 90-percentiles are shown. Unit: µg C m-3.